# Nanoscale architecture of the *Schizosaccharomyces pombe* contractile ring

**Nathan A McDonald[1], Abigail L Lind[2], Sarah E Smith[3], Rong Li[4], Kathleen L Gould[1]\***

[1]Department of Cell and Developmental Biology, Vanderbilt University, Nashville, United States; [2]Department of Biomedical Informatics, Vanderbilt University Medical Center, Nashville, United States; [3]Stowers Institute for Medical Research, Kansas City, United States; [4]Department of Cell Biology, Johns Hopkins University School of Medicine, Baltimore, United States

**Abstract** The contractile ring is a complex molecular apparatus which physically divides many eukaryotic cells. Despite knowledge of its protein composition, the molecular architecture of the ring is not known. Here we have applied super-resolution microscopy and FRET to determine the nanoscale spatial organization of *Schizosaccharomyces pombe* contractile ring components relative to the plasma membrane. Similar to other membrane-tethered actin structures, we find proteins localize in specific layers relative to the membrane. The most membrane-proximal layer (0–80 nm) is composed of membrane-binding scaffolds, formin, and the tail of the essential myosin-II. An intermediate layer (80–160 nm) consists of a network of cytokinesis accessory proteins as well as multiple signaling components which influence cell division. Farthest from the membrane (160–350 nm) we find F-actin, the motor domains of myosins, and a major F-actin crosslinker. Circumferentially within the ring, multiple proteins proximal to the membrane form clusters of different sizes, while components farther from the membrane are uniformly distributed. This comprehensive organizational map provides a framework for understanding contractile ring function.

DOI: https://doi.org/10.7554/eLife.28865.001

**\*For correspondence:**
kathy.gould@vanderbilt.edu

**Competing interests:** The authors declare that no competing interests exist.

## Introduction

In many eukaryotes, including animals and fungi, cell division is accomplished by an actin- and myosin-based contractile apparatus (*Green et al., 2012*). This complex structure is tightly linked to the plasma membrane and uses myosin motors to constrict an F-actin ring (*Cheffings et al., 2016*), pulling opposing sides of the membrane together.

Studies in *S. pombe* have led the cytokinesis field in identifying components and regulators of the contractile ring (*Cheffings et al., 2016*; *Pollard and Wu, 2010*; *Rincon and Paoletti, 2016*; *Willet et al., 2015a*; *Goyal et al., 2011*). Genetic screens as well as genome-wide and targeted localization studies have determined a complete 'parts list' of protein components that comprise the ring, many of which are conserved in higher eukaryotes (*Nurse et al., 1976*; *Balasubramanian et al., 1998*; *Chang et al., 1996*; *Matsuyama et al., 2006*). 318 proteins are annotated as localizing to the *S. pombe* division site (*Matsuyama et al., 2006*; *Wood et al., 2012*), which includes both the contractile ring and the lining of the division septum formed during ring constriction. Only a subset of these proteins (38 according to PomBase annotation [*Wood et al., 2012*]) make up the contractile ring itself.

Though the proteins that comprise the contractile ring have been identified, how these components are knit together into a functional division machine remains unclear despite several substantive efforts towards unraveling this complex question. The *S. pombe* contractile ring forms in the middle of the cell from precursor 'nodes', membrane-tethered protein foci that contain anillin Mid1, IQGAP Rng2, myosin-II heavy chain Myo2 and light chains Cdc4 and Rlc1, F-BAR Cdc15, and formin Cdc12 (*Wu et al., 2006*). Precursor nodes coalesce into a contiguous ring over ~20 min that recruits many additional components over a further ~20 min before constriction after mitotic exit. The orientation of 5 components within these nodes has been determined (*Laporte et al., 2011*), and quantitative fluorescence studies have even been used to estimate the number of molecules of many proteins per node as well as in the fully-formed ring (*Laporte et al., 2011*; *Wu and Pollard, 2005*). Knowledge from these studies has been incorporated into mathematical models which attempt to understand ring formation and constriction. A search-capture-pull-release model of node condensation was found to recapitulate basic ring formation (*Vavylonis et al., 2008*; *Ojkic et al., 2011*), while biophysical tension measurements of the ring have been used to model ring constriction (*Stachowiak et al., 2014*). Though these models are becoming increasingly complex and explanatory, the field is hampered by sparse information about the fundamental molecular architecture of the ring.

Ultimately, the resolution limit of conventional fluorescence microscopy (~250 nm) restrains the spatial information attainable by studies of *S. pombe* nodes and contractile rings, each only 100–200 nm in width (*Wu et al., 2006*; *Laplante et al., 2016*). At higher resolution, one electron microscopy study revealed that the ring is composed of a dense array of 1000–2000 F-actin filaments with mixed directionality (*Kamasaki et al., 2007*); however, additional protein components could not be detected with this technique. New super-resolution microscopy technologies, based on the precise (<50 nm) localization of single photoactivated fluorescent molecules (*Betzig et al., 2006*; *Rust et al., 2006*; *Hess et al., 2006*), have the potential to drive our understanding of the contractile ring to a truly molecular level.

Super-resolution methods have recently been effective at determining the molecular architecture and revealing the inner mechanics of multiple cellular structures (*Sydor et al., 2015*). In focal adhesions, the plasma membrane and F-actin were found to be separated by distinct layers of proteins: an integrin signaling layer, a force transduction layer, and an actin regulatory layer (*Kanchanawong et al., 2010*), revealing a potential mechanism of force-induced focal adhesion formation and maintenance. At centrosomes, the pericentriolar matrix (PCM) was found to organize into two structural layers: one directly apposed to the centriole wall and a second extending radially outward (*Mennella et al., 2012*), scaffolded by the N- and C-termini of pericentrin-like protein, respectively. Furthermore, super-resolution microscopy has revealed previously unresolvable structures, such as actin-spectrin periodic repeats which coat the membranes of axons in animals (*Xu et al., 2013*; *He et al., 2016*). Super-resolution microscopy has also recently been applied to *S. pombe* cytokinesis in a study that investigated 6 proteins' orientations within precursor nodes relative to each other (*Laplante et al., 2016*).

Using fluorescence photoactivation localization microscopy (fPALM) we approached the question of the molecular architecture of the contractile ring by comprehensively mapping 29 protein components' spatial organization relative to the underlying plasma membrane. We determined that the contractile ring is composed of layers of protein components at distinct positions interior to the plasma membrane, similar to other membrane-tethered actin structures like focal adhesions (*Kanchanawong et al., 2010*) and cadherin junctions (*Bertocchi et al., 2017*), a conclusion verified by fluorescence resonance energy transfer (FRET) experiments. Moreover, we find certain components are uniformly distributed circumferentially within the ring, while others form clusters of various sizes and variable spacing. These data provide a structural framework for understanding the formation, mechanics, and regulation of the contractile ring.

# Results

## fPALM strategy for determining the spatial organization of contractile ring components

To determine the molecular architecture of the contractile ring using fPALM, we measured the precise spatial distribution of 29 proteins endogenously tagged with mMaple3 relative to the plasma

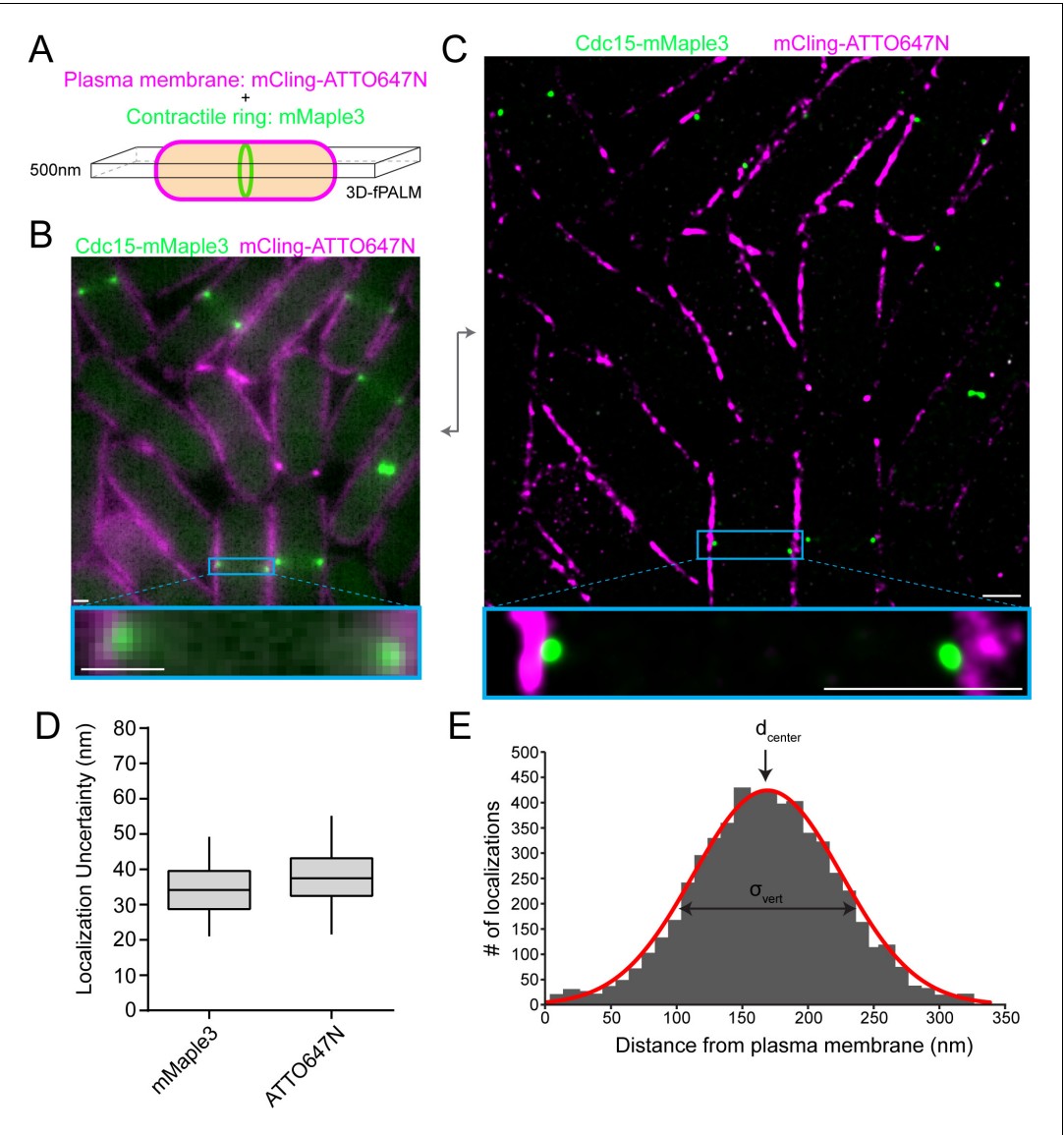

**Figure 1.** fPALM strategy to localize contractile ring proteins relative to the plasma membrane with nanometer resolution. (**A**) Schematic of fPALM sample setup. Contractile ring proteins were endogenously tagged with mMaple3 while the plasma membrane was labeled with mCling-ATTO647N. A 3D-fPALM system was used to restrict the imaging plane to a narrow 500 nm Z section. (**B**) Conventional fluorescence microscopy images of a contractile ring protein and membrane dye. (**C**) Resulting 2 color fPALM image after localization analysis. Particles are visualized as normalized Gaussians with standard deviation = localization uncertainty. (**D**) Localization uncertainty achieved with mMaple3 and ATTO647N fluorophores for all data in this study. (**E**) Example histogram of Cdc15-mMaple3 localization distances from the mCling-ATTO647N membrane edge. A Gaussian curve was fitted to the data from which $d_{center}$ and $\sigma_{vert}$ parameters can be extracted. Scale bars = 1 µm.

DOI: https://doi.org/10.7554/eLife.28865.002

The following figure supplement is available for figure 1:

**Figure supplement 1.** The mCling-ATTO674N plasma membrane marker accurately identifies the plasma membrane edge.
DOI: https://doi.org/10.7554/eLife.28865.003

membrane marked with the membrane-binding probe mCling-ATTO647N (*Revelo et al., 2014*) (*Figure 1A*). The photoactivatable mMaple3 fluorophore was chosen because it has multiple advantages over the commonly used mEos3.2 (*Zhang et al., 2012*), most importantly a faster maturation time (49 versus 330 min) (*Wang et al., 2014*). The mMaple3 fluorophore is ~5 nm in size and was connected to protein termini with a short 11 amino acid linker, minimizing positional uncertainty. In many cases, the spatial distribution of both protein termini was interrogated to determine molecular orientations. We imaged the middle focal plane of cells and utilized optical astigmatism to precisely measure the Z positions of each photoactivated molecule in order to restrict our analysis to a 500 nm Z section (*Huang et al., 2008a*). This imaging scheme resulted in a significant increase in resolution from conventional fluorescence (*Figure 1B*) to fPALM images (*Figure 1C*). Overall, we achieved an average of 34 and 37 nm localization uncertainty for mMaple3 and ATTO647N fluorophores, respectively (*Figure 1D*).

Before examining contractile ring proteins, we evaluated the mCling-ATTO647N probe's ability to identify the plasma membrane. First, we verified that mCling-ATTO647N colocalizes with the FM4-64 membrane dye (*Figure 1—figure supplement 1A*). In fact, mCling is similarly trafficked into intracellular membrane compartments, confirming its membrane incorporation (*Figure 1—figure supplement 1A*). However, at super-resolution the mCling-ATTO647N signal is wider than the ~10 nm plasma membrane (*Figure 1—figure supplement 1B*), indicating that mCling-ATTO647N may also bind nonspecifically to the cell wall. Nevertheless, the interior edge of the mCling signal aligns extremely well with the edge of an Acyl-mMaple3 signal, validating the use of mCling to mark the cytosolic face of the plasma membrane (blue line, *Figure 1—figure supplement 1B*).

To quantify the spatial distribution of each protein component relative to the plasma membrane, we performed an analysis similar to that performed on focal adhesion and cadherin junction proteins (*Kanchanawong et al., 2010*; *Bertocchi et al., 2017*). The cytoplasmic edge of the plasma membrane was defined in an unbiased manner as the position where the mCling-ATTO647N signal dropped to 5%. The distance from the defined plasma membrane edge was then calculated for each individual protein localization in a contractile ring side-view 'spot'. The average distance from the membrane ($d_{center}$), a vertical width parameter ($\sigma_{vert}$), and a horizontal width parameter ($\sigma_{width}$) were determined as in Kanchanawong et al. (*Kanchanawong et al., 2010*), and these values were then averaged across multiple rings. We restricted our analysis to fully formed, unconstricted contractile rings that are present in cells for 15–20 min. Cells containing rings of this stage were identified based on the absence of precursor nodes, the presence of a tightly concentrated ring absolutely perpendicular to the cell's long axis, and the lack of any membrane ingression or septum formation (*Figure 1—figure supplement 1C*).

## Spatial distribution of structural contractile ring components

We first determined the spatial distribution of 19 structural components of the contractile ring relative to the membrane (*Figure 2A–B*, *Figure 2—figure supplement 1*). Endogenous mMaple3 fusions of these proteins were functional and had no impact on growth or division (*Figure 2—figure supplement 2*). The resolution of the mMaple3 probe (*Figure 1D*) was sufficient to distinguish differences in distance from the membrane between components. Overall, we found the fully-formed contractile ring is made up of components 182 ± 26 nm wide and extends from the plasma membrane 293 ± 64 nm into the cytoplasm.

Multiple membrane-binding proteins including the anillin Mid1, the F-BAR proteins Cdc15 and Imp2, and the septin Spn3 are positioned within 80 nm of the plasma membrane. Both termini of Mid1 are within 50 nm of the membrane, consistent with direct membrane binding through its C2 domain (*Sun et al., 2015*; *Rincon and Paoletti, 2012*; *Celton-Morizur et al., 2004*). Cdc15 and Imp2 contain N-terminal F-BAR domains that also directly bind the membrane (*Roberts-Galbraith et al., 2010*; *McDonald et al., 2016*; *McDonald et al., 2015*). We find Spn3 near the membrane, consistent with its assembly into membrane-bound oligomers with additional septin subunits (*An et al., 2004*). In addition to membrane-binding proteins, the C-terminal tail of the essential contractile ring myosin-II Myo2 (*Kitayama et al., 1997*; *Palani et al., 2017*) is near the membrane, as was observed in cytokinesis precursor nodes (*Laporte et al., 2011*; *Laplante et al., 2016*). The N-terminus of the formin Cdc12 is also found near the membrane, in accord with direct binding to the F-BAR domain of Cdc15 (*Willet et al., 2015b*; *Carnahan and Gould, 2003*). The C-terminus of

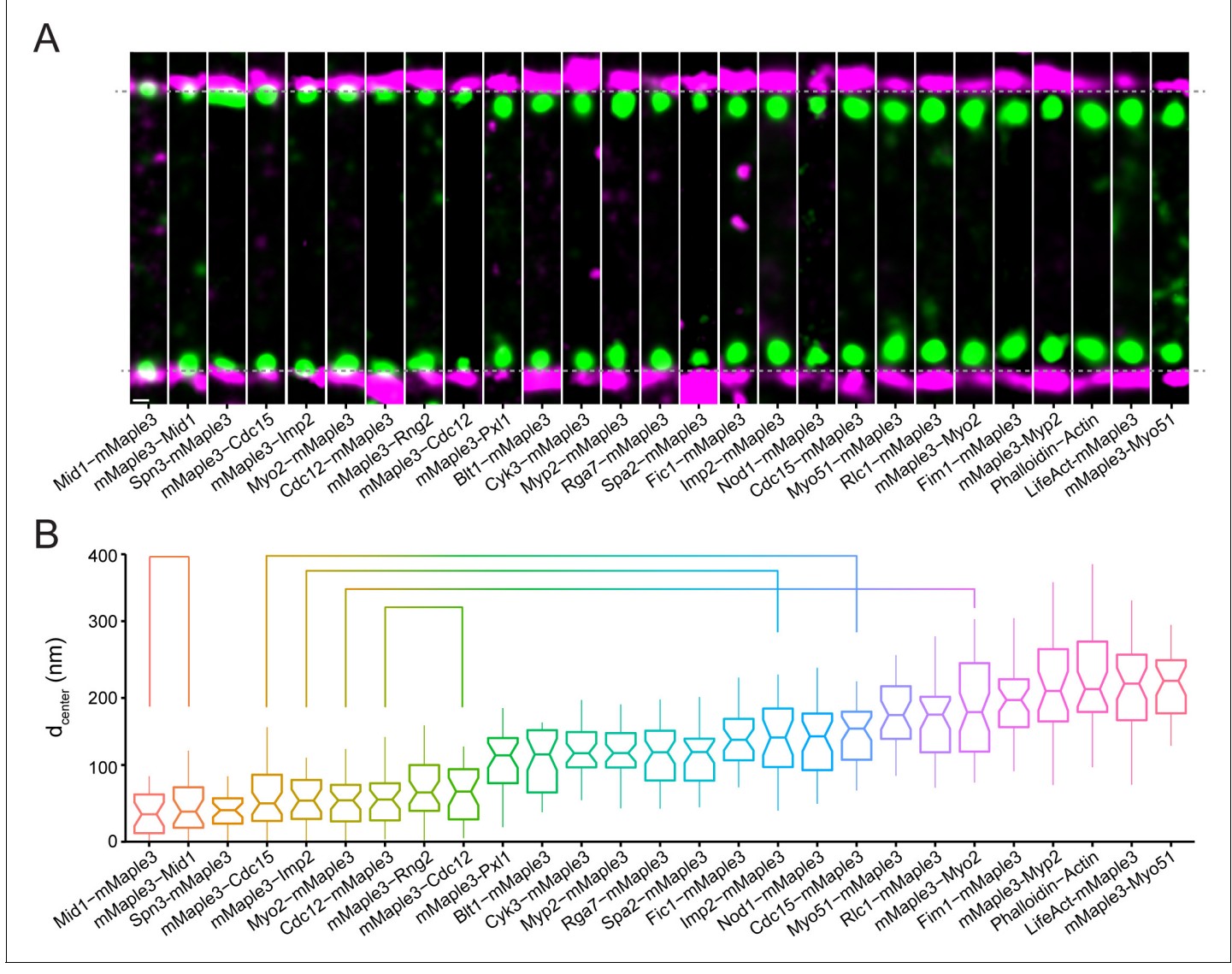

**Figure 2.** Nanoscale organization of contractile ring structural components. (**A**) Representative fPALM images of structural contractile ring components. Scale, 100 nm. Particles are visualized as normalized Gaussians with standard deviation = localization uncertainty. Dashed lines indicate plasma membrane edges. (**B**) Distance from the plasma membrane ($d_{center}$) of structural contractile ring components. Box plots depict 1st and 3rd quartiles and median; Whiskers, minimum and maximum; Notches, 95% confidence intervals. Lines connect proteins labeled on opposite termini.

DOI: https://doi.org/10.7554/eLife.28865.004

The following figure supplements are available for figure 2:

**Figure supplement 1.** Vertical and horizontal width parameters ($\sigma_{vert}$ and $\sigma_{width}$) and localization values for strains in *Figure 2*.

DOI: https://doi.org/10.7554/eLife.28865.005

**Figure supplement 2.** Endogenous mMaple3 tags do not perturb normal growth and division.

DOI: https://doi.org/10.7554/eLife.28865.006

Cdc12 is found at a similar distance, further supporting its localization to a membrane-proximal stratum.

Slightly further from the membrane (80–160 nm) we find the IQGAP Rng2, a scaffolding protein critical for contractile ring formation and constriction (*Padmanabhan et al., 2011*; *Tebbs and Pollard, 2013*). Multiple accessory proteins important for ring integrity are also present in this layer and are linked to the underlying layer through direct protein-protein interactions: Cyk3, Spa2, Blt1, paxillin Pxl1, Nod1, and Fic1 (*Roberts-Galbraith et al., 2010*; *Bohnert and Gould, 2012*; *Ren et al., 2015*; *Pollard et al., 2012*; *Zhu et al., 2013*; *Guzman-Vendrell et al., 2013*; *Pinar et al., 2008*;

*Ge and Balasubramanian, 2008*). Pxl1's localization to an intermediate layer between the membrane and F-actin is comparable to paxillin's position between membrane-embedded integrins and F-actin in focal adhesions (*Kanchanawong et al., 2010*). At similar intermediate distances from the membrane, we find the tail domains of two additional myosins that contribute to *S. pombe* cytokinesis: the non-essential myosin-II, Myp2 (*Bezanilla et al., 1997*; *Mulvihill and Hyams, 2003*), as well as a myosin-V, Myo51 (*Win et al., 2001*; *Tang et al., 2016*). Though both are non-essential, Myp2 is implicated in ring constriction (*Palani et al., 2017*; *Laplante et al., 2015*) while Myo51 contributes to ring formation and constriction (*Tang et al., 2016*).

Interestingly, we find that the C-termini of three F-BAR proteins (Cdc15, Imp2, and Rga7) extend away from their membrane-bound F-BAR domains into the intermediate layer of the ring (lines, *Figure 2B*). These proteins contain central regions predicted to be unstructured, which may permit the SH3 domains of Cdc15 and Imp2 to extend inward ~100 nm distances and connect with their multiple interactors also found at this layer including Pxl1, Spa2, Fic1, Cyk3 (*Figure 2*), and Rgf3 (*Figure 3*) (*Roberts-Galbraith et al., 2010*; *Ren et al., 2015*; *Roberts-Galbraith et al., 2009*).

At the farthest distances from the membrane (160–350 nm), we find F-actin, the motor domains of the type II and type V myosins Myo2, Myp2, and Myo51, as well as Fim1, an F-actin crosslinker (*Figure 2*). Since actin is not functional as a fluorescent fusion in *S. pombe* (*Wu and Pollard, 2005*), we imaged F-actin with two probes: LifeAct-mMaple3 and Phalloidin-Alexa488. Measurements using these two methods were consistent, placing F-actin's $d_{center}$ at 219 ± 64 (LifeAct-mMaple3) or 228 ± 75 nm (Phalloidin; *Figure 2—figure supplement 1B*) from the membrane. The N-terminal head domains of Myo2, Myp2, and Myo51 are level with F-actin, while the light chain Rlc1 is ~30 nm closer to the membrane, in accord with its binding to the neck domains of myosin motors that are immediately proximal to the head domains (*Naqvi et al., 2000*). Two actin crosslinking proteins are present in the contractile ring, fimbrin Fim1 and α-actinin Ain1 (*Wu et al., 2001*), with Ain1 being the predominant cytokinetic crosslinker (*Wu et al., 2001*; *Li et al., 2016*). We were unable to produce a functional mMaple3-Ain1, but anticipate it would localize in the same region as F-actin, similar to Fim1.

## Spatial distribution of contractile ring signaling components

The contractile ring contains a plethora of signaling proteins that influence the behavior of its structural elements including kinases, phosphatases, GTPase exchange factors (GEFs) and GTPase activating proteins (GAPs) (*Wood et al., 2012*; *Bohnert and Gould, 2011*). We examined the spatial distribution of 10 such components that are present in fully formed unconstricted rings (*Figure 3A–B*, *Figure 3—figure supplement 1–2*). The kinases Orb2/Pak1/Shk1, Sid2, and Pck1 are relatively close to the membrane, on average <110 nm. In focal adhesions, regulators such as Focal Adhesion Kinase (FAK) are also found at an intermediate layer between the membrane and F-actin (*Kanchanawong et al., 2010*). In contrast, the N-terminus of the kinase Pom1 was found farther away from the membrane (149 nm) despite the presence of a nearby membrane-binding motif (*Hachet et al., 2011*). It may be that Pom1 localization to the contractile ring utilizes a different mechanism than when targeted to the cell tip cortex (*Hachet et al., 2011*). It is noteworthy that Pom1 contains multiple PxxP motifs and may bind the SH3 network established by Cdc15 and Imp2. The RhoGEF Rgf3, a known partner of Cdc15 and Imp2 SH3 domains (*Ren et al., 2015*), is also present at an intermediate height.

Phosphatases Clp1 (Cdc14-related) and Ppb1 (Calcineurin), RhoGAP Rga2, and RhoGEF Gef2 were also found in an intermediate stratum. Gef2 is in complex with Nod1 (*Zhu et al., 2013*) and is found at a similar elevation to its binding partner, 145 nm and 140 nm, respectively. Rad24, a 14-3-3 protein, while not technically a signaling molecule, binds to many phosphoproteins to control signaling pathways (*Mackintosh, 2004*; *Aitken, 2006*). We find the majority of Rad24 high in elevation, similar to F-actin, though it also localizes throughout the cytoplasm.

## Distinguishing contractile ring layers with dual fluorophores

To corroborate the relative positions of proteins determined individually, we next imaged multiple strains containing the mMaple3 tag on two separate proteins. First, we imaged two proteins in the same layer: mMaple3-Cdc15 and Mid1-mMaple3 in the proximal layer or Cdc15-mMaple3 and Imp2-mMaple3 in the intermediate layer. Confirming their overlapping localization, distributions

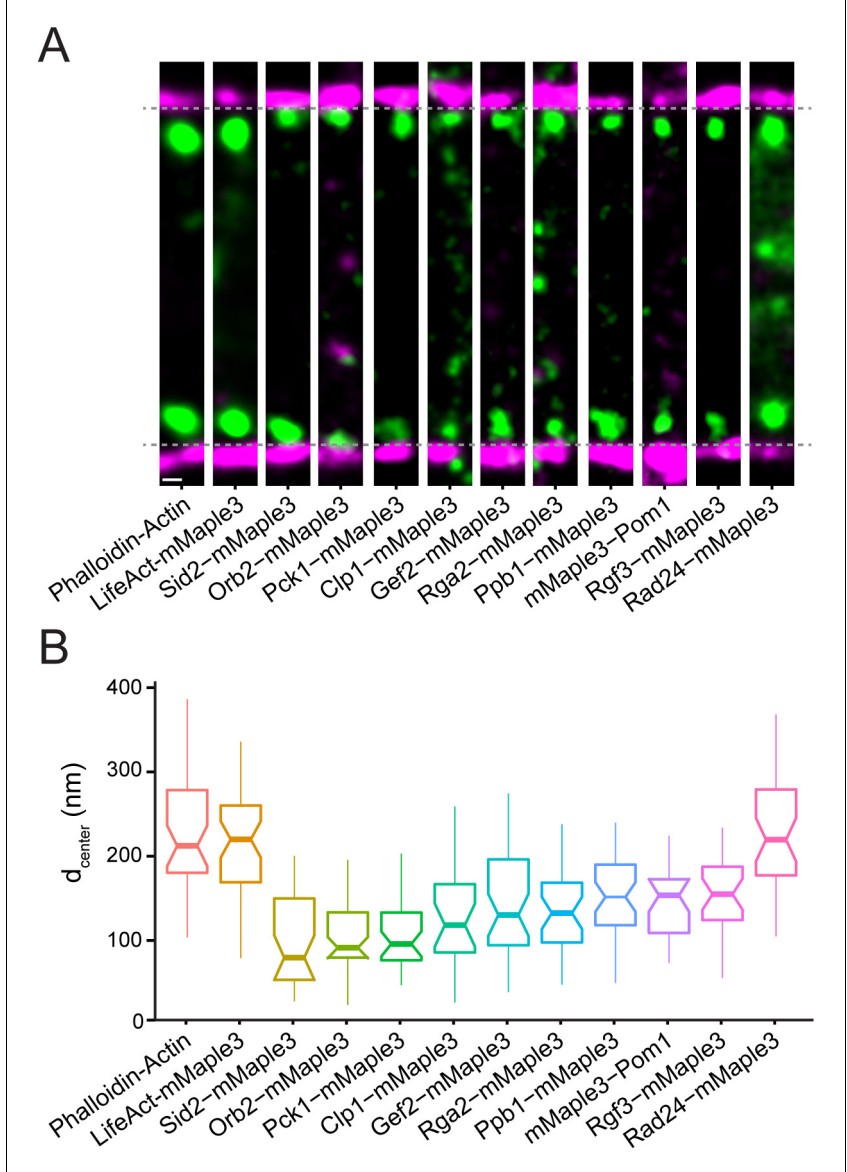

**Figure 3.** Nanoscale organization of contractile ring signaling components. (**A**) Representative fPALM images of signaling contractile ring components. Scale, 100 nm. Particles are visualized as normalized Gaussians with standard deviation = localization uncertainty. Dashed lines indicate plasma membrane edges. (**B**) Distance from the plasma membrane ($d_{center}$) of signaling contractile ring components. Box plots depict 1st and 3rd quartiles and median; Whiskers, minimum and maximum; Notches, 95% confidence intervals.

DOI: https://doi.org/10.7554/eLife.28865.007

The following figure supplements are available for figure 3:

**Figure supplement 1.** Vertical and horizontal width parameters ($\sigma_{vert}$ and $\sigma_{width}$) and localization values for strains in *Figure 3*.

DOI: https://doi.org/10.7554/eLife.28865.008

**Figure supplement 2.** Endogenous mMaple3 tags do not perturb normal growth and division.

DOI: https://doi.org/10.7554/eLife.28865.009

relative to the plasma membrane remained single narrow peaks (*Figure 4A–B*, *Figure 4—figure supplement 1*). Next, we combined tagged proteins from adjacent layers: mMaple3-Cdc15 and Imp2-mMaple3, as well as mMaple3-Cdc15-mMaple3 with tags on its two termini in proximal and intermediate layers. In these cases, a single peak was still observed, but the peak was significantly

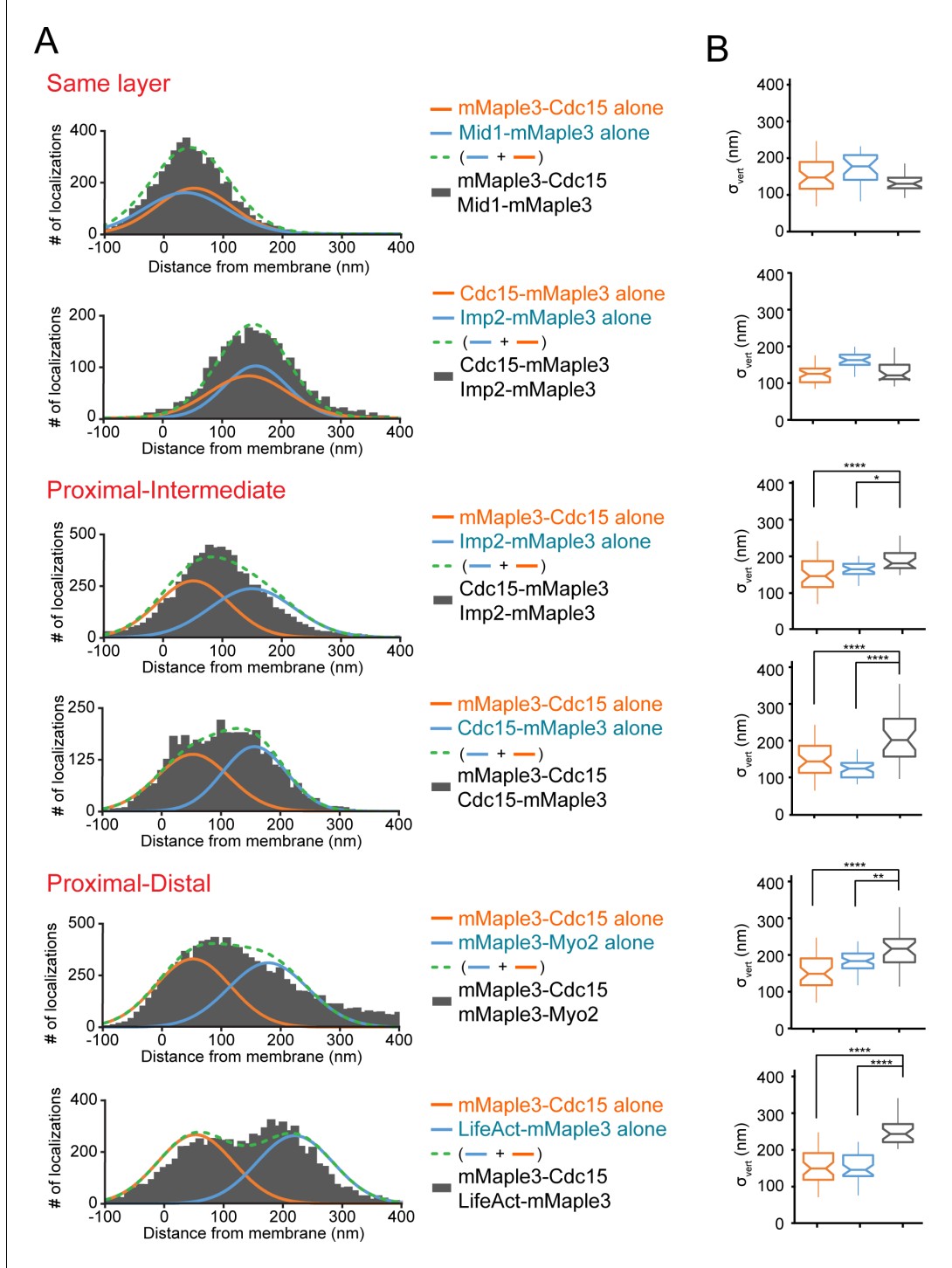

**Figure 4.** Distinguishing contractile ring layers with pairs of mMaple3-tagged proteins. (**A**) Comparison of single mMaple3-tagged proteins in similar or disparate localizations in the contractile ring. Orange and blue lines represent average localization of individual proteins from *Figures 2–3*, while dashed green lines represent the theoretical localization of two individual proteins imaged simultaneously. Grey histograms depict actual localization data from the indicated dual mMaple3-tagged strains. (**B**) Comparison of $\sigma_{vert}$ (localization spread) parameters between single and double tagged strains reveals wider peaks when two proteins are separated spatially. One way ANOVA tests were performed between the indicated samples. *$p<0.05$; **$p<0.01$; ****$p<0.0001$.

DOI: https://doi.org/10.7554/eLife.28865.010

The following figure supplement is available for figure 4:

**Figure supplement 1.** Visualizing pairs of mMaple3 tagged protein.

*Figure 4 continued on next page*

Figure 4 continued

DOI: https://doi.org/10.7554/eLife.28865.011

wider, matching predictions (green dashed lines, *Figure 4A*). Finally, we examined strains containing proteins in proximal and distal layers: mMaple3-Cdc15 and mMaple3-Myo2, along with mMaple3-Cdc15 and LifeAct-mMaple3. In these cases, either an even wider peak or two resolvable peaks were observed, confirming the spatial separation of proximal and distal components (*Figure 4A–B*).

## Distinct layers around F-BAR protein scaffolds

The fPALM analysis of Cdc15 and Imp2 F-BAR proteins indicated that their C-termini were positioned in a different stratum of the contractile ring than their N-terminal membrane-bound F-BAR domains. To verify the existence of distinct layers of F-BAR- and SH3-scaffolded components with a different approach, we placed a mCherry fluorophore on either the N-terminus (near the F-BAR domain) or C-terminus (near the SH3 domain) of Cdc15 in combination with GFP- or mNeonGreen (mNG)-conjugated ring components and analyzed each strain by fluorescent resonance energy transfer (FRET) (*Figure 5A–B*, *Figure 5—figure supplement 1A*).

Supporting the hypothesis that Cdc15 and probably other F-BARs scaffold distinct layers of proteins in the contractile ring, we detected specific FRET signals between mCherry-Cdc15 and membrane-bound Acyl-GFP, GFP-Imp2, GFP-Rga7, and both mNG-Cdc12 and Cdc12-mNG (*Figure 5A*). A strong FRET signal between mCherry-Cdc15 and Acyl-GFP is consistent with the binding of Cdc15's F-BAR to anionic phospholipids within the plasma membrane (*Roberts-Galbraith et al., 2010*; *McDonald et al., 2015*). Furthermore, the FRET signal of mCherry-Cdc15 with two other F-BAR domains (GFP-Imp2 and GFP-Rga7) indicates that all three cytokinesis F-BARs are packed in close proximity upon the membrane, despite the fact that they do not heterodimerize (*Roberts-Galbraith et al., 2009*; *Martín-García et al., 2014*). The FRET signal of mCherry-Cdc15 with both termini of Cdc12 is consistent with the membrane-proximal spatial distribution of both Cdc12 termini observed by fPALM analysis. Importantly, the FRET signals of membrane-proximal ring components with mCherry-Cdc15 were specific, as these particular GFP- and mNG-tagged components did not display significant FRET with Cdc15 tagged at its C-terminus with mCherry (*Figure 5C–D*).

Cdc15-mCherry, on the other hand, showed significant FRET signals with Imp2-GFP, Rga7-GFP, and five known binding partners of its SH3 domain: Fic1-GFP, Pxl1-GFP, GFP-Spa2, GFP-Rgf3, and GFP-Cyk3 (*Figure 5B*) (*Roberts-Galbraith et al., 2010*; *Ren et al., 2015*; *Roberts-Galbraith et al., 2009*). The detection of FRET between Cdc15-mCherry and the C-terminal domains of Imp2 and Rga7 supports the spatial distribution of these molecules proposed from our fPALM analysis and indicates these domains, like their F-BAR domains, are positioned close enough together to support FRET. The detection of FRET signals between known Cdc15 SH3 domain-binding partners and Cdc15-mCherry was expected. Moreover, the specificity of the signal for Cdc15-mCherry over mCherry-Cdc15 (*Figure 5—figure supplement 1B*) supports the extended orientation of Cdc15 proposed from our fPALM analysis. We detected no significant FRET signal between mCherry-labeled Cdc15 and multiple additional contractile ring proteins (*Figure 5—figure supplement 1C*), again consistent with the CR being composed of distinct strata.

## Circumferential clustering capability of contractile ring scaffolds

Our fPALM analysis of the molecular architecture of the contractile ring described above utilized a 'side-view', designed to easily measure distances from the membrane. With this view, information about how protein components organize along the circumferential axis of the ring is missing. Therefore, we performed additional fPALM imaging at the top plane of cells (*Figure 6A*) to capture a circumferential view of the contractile ring and assess the organization of a subset of contractile ring components. To quantify the level of clustering, we calculated the Inverse Difference Moment (IDM) of fPALM images along the ring circumferential axis (*Haralick et al., 1973*). IDM is a measure of local homogeneity; values close to 1 have homogenous local intensities (uniform) and those further from 1 have heterogeneous local intensities (clustered). We simulated a completely uniform and a clustered distribution for comparison (top two panels, *Figure 6B–D*).

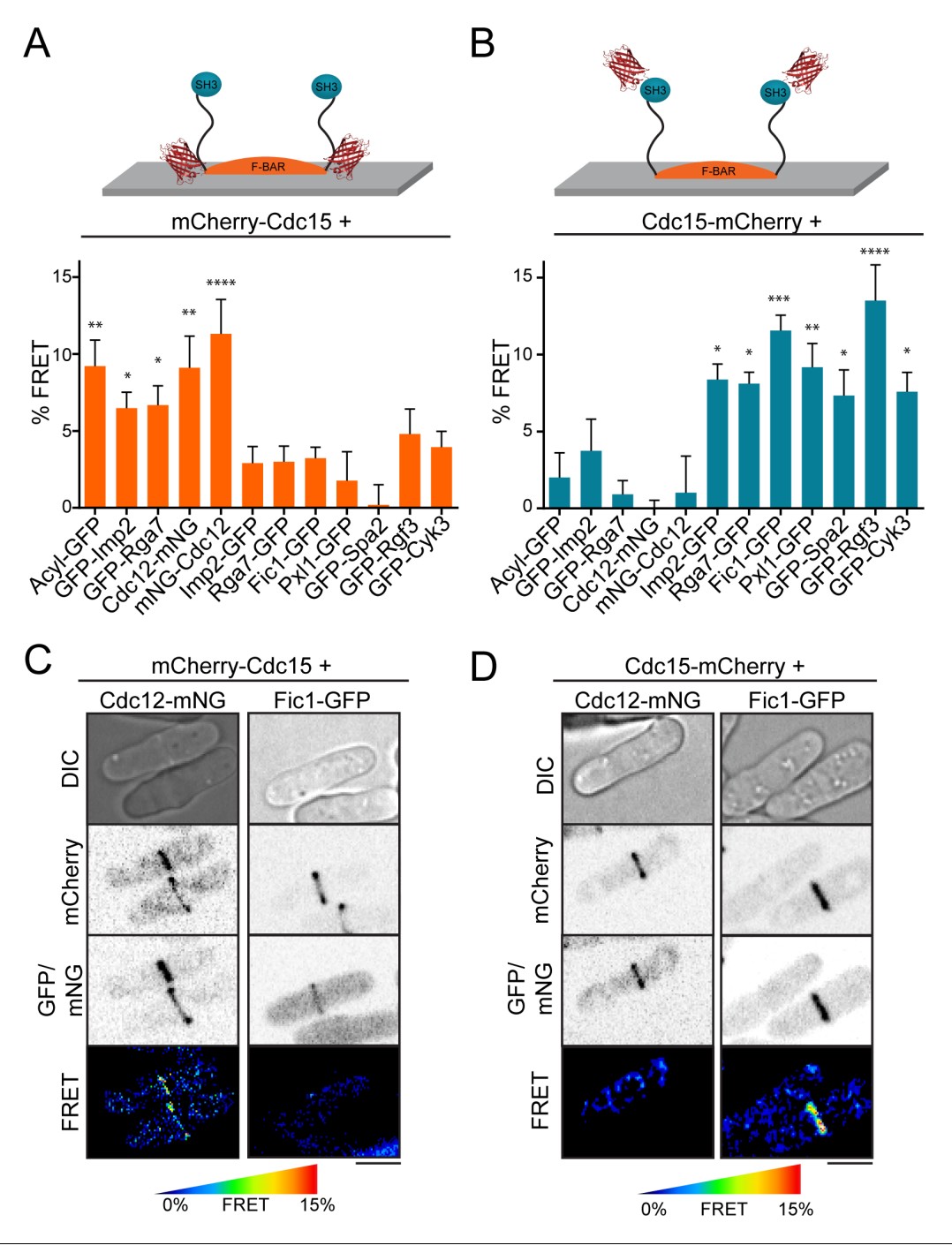

**Figure 5.** FRET confirms distinct layers around F-BAR proteins in the contractile ring. (**A–B**) Quantification of FRET between mCherry-Cdc15 (**A**) or Cdc15-mCherry (**B**) and GFP- or mNG-conjugated contractile ring components. See also *Figure 5—figure supplement 1* for direct mCherry-Cdc15:Cdc15-mCherry comparisons. % FRET is the increase in GFP donor signal following mCherry acceptor photobleaching. One way ANOVA tests were performed for each strain against an Rlc1-GFP negative control. *$p < 0.05$; **$p < 0.01$; ***$p < 0.001$; ****$p < 0.0001$. (**C–D**) Cdc12-mNG and Fic1-GFP FRET signals at the contractile ring are specific to mCherry-Cdc15 (**C**) or Cdc15-mCherry (**D**). Scale = 4 µm.

DOI: https://doi.org/10.7554/eLife.28865.012

The following figure supplement is available for figure 5:

**Figure supplement 1.** Intermolecular FRET experiments.

*Figure 5 continued on next page*

*Figure 5 continued*

DOI: https://doi.org/10.7554/eLife.28865.013

Surprisingly, we detected multiple distinct patterns of circumferential spatial distribution among the subset of contractile ring proteins examined. Some appeared uniform in distribution while others adopted variable clustered organizations; even different domains within a single molecule were distributed differently (*Figure 6B–D*). Distally, F-actin, detected by LifeAct-mMaple3, was distributed uniformly along the ring circumference, consistent with electron micrographs of the ring (*Kamasaki et al., 2007*). Also in the distal layer, the N-terminus of Myo2 was uniformly distributed. Within the intermediate layer, the C-termini of Myp2, Fic1, and Cdc15 were evenly distributed. In contrast, the domains of Cdc15 and Myo2 close to the membrane appeared to form clusters spaced at regular intervals along the circumferential axis with diameters of 71 ± 17 and 104 ± 26 nm, respectively. Also in the proximal layer, Rng2 and Mid1 were present in larger, unevenly spaced clusters 201 ± 49 and 200 ± 44 nm in diameter, respectively. Though these clustered components all reside in the proximal layer, not all proximal components were clustered. Spn3 displayed a uniform distribution which might be expected for dense septin filament bundles that are not entirely in register (*Oh and Bi, 2011*) and because septins are never present in early cytokinesis nodes (*Tasto et al., 2003*; *Berlin et al., 2003*).

Cdc15's N-terminal F-BAR forms linear oligomers (*McDonald et al., 2015*); to test if oligomerization was responsible for organization of the observed larger scale clusters, we imaged a mMaple3-Cdc15(E30K E152K) oligomerization mutant. Indeed the oligomerization mutant prevented cluster formation. Interestingly, the oligomerization mutation also resulted in a widened mMaple3-Cdc15 ring thickness, suggesting oligomerization may also help concentrate Cdc15 in the ring. In contrast, the deletion of *mid1* did not prevent Cdc15 cluster formation.

## Discussion

Using fPALM super-resolution imaging, we have determined the spatial organization of 29 central components of the contractile ring relative to the membrane. We found that these components organize into distinct connected layers rising from the plasma membrane and extending up to 350 nm towards the interior of the cell (*Figure 7* and *Video 1*). We also discovered that certain components adopt distinct spatial distributions along the circumference of the contractile ring ranging from uniform to large, irregularly sized and spaced clusters. Our measurements establish a strong foundation for a comprehensive understanding of the molecular architecture and function of a eukaryotic contractile ring.

### Structural layers within the contractile ring

Our analysis of 29 components of the contractile ring by fPALM indicates the existence of distinct layers tethering F-actin to the plasma membrane, similar to two previously investigated membrane-tethered F-actin structures, focal adhesions (*Kanchanawong et al., 2010*) and cadherin junctions (*Bertocchi et al., 2017*).

Nearest to the membrane (0–80 nm), proteins with membrane-binding domains scaffold additional components through a network of protein-protein interactions. The anillin Mid1, tethered to the membrane via its C2 domain, binds the tail of the essential type II myosin, Myo2 (*Motegi et al., 2004*), as well as extending to bind IQGAP Rng2 (*Padmanabhan et al., 2011*), which in turn stabilizes the Myo2-Mid1 interaction (*Laporte et al., 2011*). The F-BAR domain of Cdc15, also bound directly to the membrane, scaffolds the formin Cdc12 (*Willet et al., 2015b*; *Carnahan and Gould, 2003*). The membrane-binding F-BAR domain of Rga7 may also have a ring binding partner, Rng10 (*Liu et al., 2016*), suggesting that F-BAR domains may scaffold additional ring components, linking their partners indirectly but closely to the membrane. It is not surprising that the integrity of this proximal layer of the contractile ring is critical for strong attachment to the plasma membrane. In fact, additional time is required to assemble the ring when Mid1 is missing (*Huang et al., 2008b*; *Hachet and Simanis, 2008*; *Wu et al., 2003*), while rings can slide and disassemble when Cdc15 abundance is reduced (*McDonald et al., 2015*; *Arasada and Pollard, 2014*). It is surprising that the ring remains associated with the plasma membrane in the majority of cells lacking Mid1 or fully

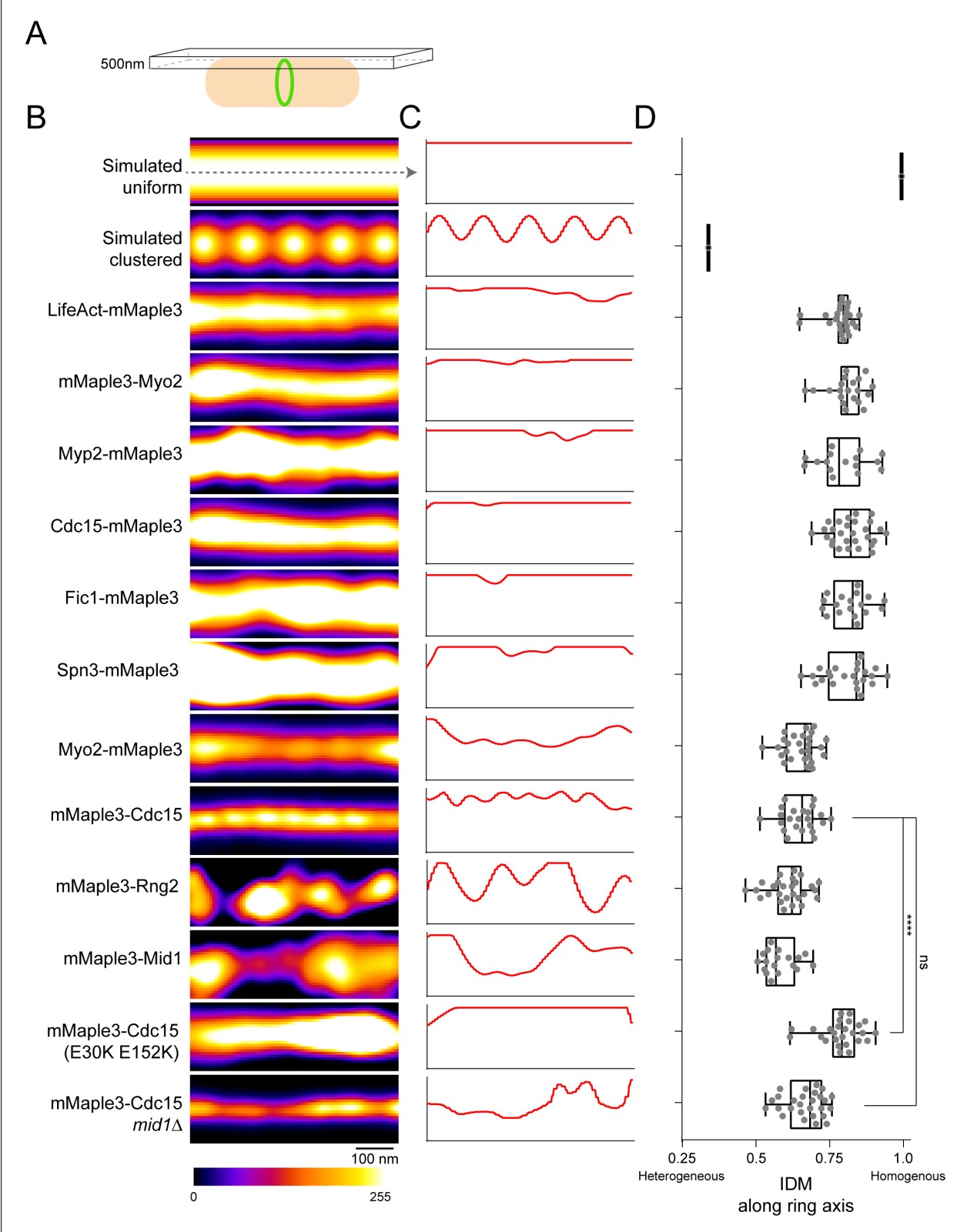

**Figure 6.** Lateral organization of components in the contractile ring. (A) Schematic of fPALM setup with Z focus at the top of cells. (B) Representative fPALM image of the top of contractile rings with the indicated components labeled. Simulated clustered or smooth ring examples are included for comparison. (C) Linescans through the center of images in (A). (C) Quantification of the Inverse Difference Moment (IDM) of a grey level co-occurance

*Figure 6 continued on next page*

*Figure 6 continued*

matrix along the contractile ring's circumferential axis. IDM is a measure of local homogeneity, see Materials and methods. Lines indicate a one way ANOVA performed against mMaple3-Cdc15; ****p<0.0001.

DOI: https://doi.org/10.7554/eLife.28865.014

functional Cdc15 or both (*McDonald et al., 2015*), indicating that other ring components important for establishing this tight plasma membrane-contractile ring linkage must exist, especially during constriction when Mid1 leaves the ring (*Wu et al., 2003*).

At 80–160 nm away from the membrane, we find the C-termini of Cdc15 and Imp2, which contain their SH3 domains. These domains are functionally redundant (*Ren et al., 2015*), but having one or the other is essential for cytokinesis and cell viability (*Roberts-Galbraith et al., 2009*), demonstrating the importance of an SH3-scaffolded network within the contractile ring. Multiple binding partners of Cdc15's and Imp2's SH3 domains including Pxl1, Fic1, Spa2, Rgf3, and Cyk3 (*Roberts-Galbraith et al., 2010*; *Ren et al., 2015*; *Roberts-Galbraith et al., 2009*) concentrate in this intermediate zone. Cyk3 itself contains an SH3 domain, likely further strengthening this interaction network. Cdc15 and Imp2 SH3 domains bind a K/RxLPxΦP motif that is found in a number of additional contractile ring proteins (*Ren et al., 2015*) that may also be present in this zone. It is interesting to note that the functions of the majority of these proteins remains unclear, and that their functions are not strictly essential. However, given the multiple protein interaction domains present in these components and their spatial organization, we speculate that eliminating pairs or subsets of these elements would undermine the networking of this zone. This may result in weakening the linkage between the strata of the contractile ring ultimately leading to loss of ring integrity and cell viability, irrespective of any mechanistic contributions of these components to ring function. In support of this

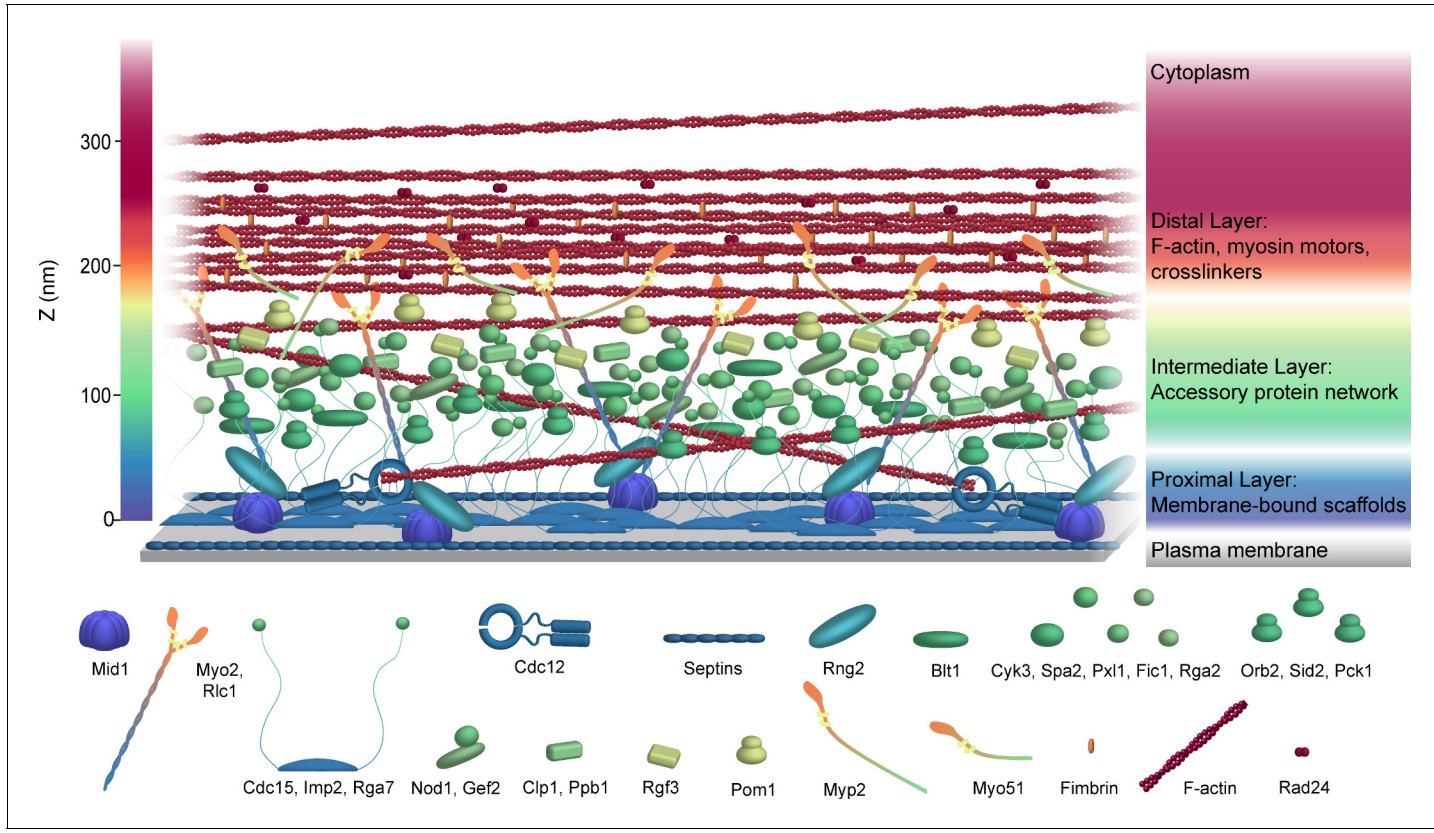

**Figure 7.** Schematic model of the *S. pombe* contractile ring molecular architecture. Depicted protein positions were calculated experimentally and are color-coded in the Z dimension. Note that this model does not incorporate stoichiometry. See also *Video 1*.

DOI: https://doi.org/10.7554/eLife.28865.015

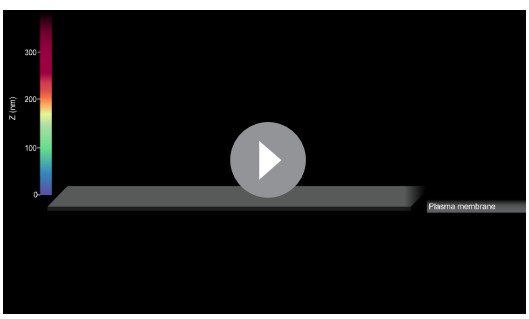

**Video 1.** Animated schematic model of the *S. pombe* contractile ring molecular architecture. Depicted protein positions were calculated experimentally and are color-coded in the Z dimension. Note that this model does not incorporate stoichiometry.
DOI: https://doi.org/10.7554/eLife.28865.016

hypothesis, synthetic lethality has been observed in several cases when null alleles of these genes were combined (*Ren et al., 2015*; *Pinar et al., 2008*; *Roberts-Galbraith et al., 2009*; *Liu et al., 2016*; *Chen et al., 2016*). Indeed in the complete absence of Cdc15 and its SH3 interaction network, F-actin rings can still form but quickly disintegrate (*Laporte et al., 2011*), underscoring the stabilizing function of the intermediate zone.

Farthest from the membrane (160–350 nm), we find the center of F-actin localization. The elevated localization and ~200 nm thickness of F-actin matches the limited electron micrographs of *S. pombe* rings that are available (*Kamasaki et al., 2007*). It is notable that classical electron microscopy studies of animal cells have also measured the contractile ring as 250–500 nm thick vertically from the membrane (*Schroeder, 1973*). Interestingly, both termini of the formin Cdc12, which nucleates and elongates F-actin in the ring de novo, are significantly closer to the membrane. This localization suggests that F-actin may be nucleated and polymerized by Cdc12 close to the membrane, but filaments are pulled upward and incorporated into the main F-actin density by tension from myosin motors and actin crosslinkers. Indeed we found the head domains of all 3 contractile ring myosins and the actin crosslinker Fim1 at a high elevation apposed to F-actin. The essential myosin-II Myo2 is oriented with its tail close to the membrane and motor extended, an orientation also seen in precursor nodes (*Laporte et al., 2011*; *Laplante et al., 2016*). This implies a different organization than the canonical bipolar filaments that are observed in sarcomeres and mammalian cleavage furrows (*Fenix et al., 2016*; *Henson et al., 2017*; *Pollard et al., 2017*). Though bipolar filaments are not formed, an orientation with Myo2's tail bound near the membrane would provide a critical anchor for force generation by its distal motor domain. Myo2 has additionally been implicated in ring formation independent of its motor activity (*Palani et al., 2017*), a function that may be contained within domains near the membrane and other early ring components. We find the tails of the non-essential myosins Myp2 and Myo51 at an intermediate layer. Myp2 and Myo51 are single-headed myosin motors (*Tang et al., 2016*; *Bezanilla and Pollard, 2000*); embedding their tail domains in the intermediate layer may also aid in force generation upon the distal F-actin. It is likely that other components such as α-actinin and tropomyosin that depend on F-actin for ring localization are also present in the most interior layer.

We also quantified the geometry of each ring component within the ring, both in terms of ring width and in terms of strata width, e.g. spread from the center of the ring and distance from the membrane. Relative to the membrane, ring components tended to organize into similarly 'thick' strata of ~150–200 nm. In terms of width within the ring, components organized in widths between 120–220 nm, with Fim1-mMaple3 being widest and mMaple3-Imp2 being thinnest, though in general widths did not significantly correlate with components' distance from the membrane. Clearly, the membrane proximal layer is no wider than the actin layer or vice versa.

## Signaling within the contractile ring

The nanoscale spatial organization of signaling components of the contractile ring has not been studied previously, though we estimate that there are ~10 protein kinases, ~3 phosphatases, and >20 GTPases and associated GAPs and GEFs in the contractile ring. Though some of these modulators have known binding partners that may help recruit them to the appropriate layer (e.g. the protein kinase Sid2 by Blt1 (*Goss et al., 2014*), the GTPase exchange factor Gef2 by Nod1 (*Zhu et al., 2013*), and the Rho1-exchange factor Rgf3 by Cdc15 and Imp2 [*Ren et al., 2015*]), it is also possible that these enzymes accumulate in distinct zones due to more transient interactions with their substrates. We did find that most localized in the correct areas for modification of their reported substrates. Sid2 is localized at a low elevation (88 nm), properly placed to access its substrates Cdc12 (*Bohnert et al., 2013*), Spa2 (*Gupta et al., 2013*), and Clp1 (*Chen et al., 2008*). The

protein kinase Pom1 is in the intermediate layer (175 nm), overlapping with the localization of 3 known substrates: Imp2, Cdc15, and Rga7 (*Kettenbach et al., 2015*). Furthermore, the Clp1 phosphatase is present at a low-to-intermediate level (136 nm), near its scaffold Mid1 (*Clifford et al., 2008*) and known and predicted ring substrates Cdc15, Blt1, Rng2, Cyk3, Myp2, and Orb2 (*Clifford et al., 2008*; *Chen et al., 2013*).

## Protein clusters in the contractile ring

When we examined a circumferential view of the ring, we found that components adopted a range of spatial organizations. Mid1 and Rng2 form irregularly spaced 200 nm clusters, which resemble the size and shape of cytokinesis precursor nodes previously measured with different techniques (conventional microscopy and live cell fPALM) (*Wu et al., 2006*; *Laporte et al., 2011*). The membrane-proximal termini of Myo2 and Cdc15 form smaller 70–100 nm diameter regularly-spaced clusters along the circumference of the ring. All four of these proteins are present in precursor nodes (*Wu et al., 2006*), which have recently been suggested to be maintained in the fully developed contractile ring (*Laplante et al., 2016*). However, the very different diameter and regularity of Cdc15 and Myo2 clusters from those of Mid1 and Rng2 argues that at the least, node structures change as the ring develops. This is perhaps to be expected since Mid1, responsible for setting up precursor nodes, dissociates from the ring as it constricts (*Wu and Pollard, 2005*). Additionally, the relative stoichiometry of certain components in the ring change during constriction: Myo2 is concentrated during constriction while Cdc15, Rng2, and Cdc12 remain at a constant concentration (*Wu and Pollard, 2005*; *Wu et al., 2003*). Therefore, reorganization of the membrane-proximal level within the ring must occur. Future two-color super-resolution imaging could determine if large clusters contain combinations of components that persist from precursor nodes as well as any additional novel clustering behavior in the fully formed ring.

Interestingly, we found components that are further away from the membrane are uniform in their distribution, even compared to their opposite termini. In particular, the F-actin signal from LifeAct-mMaple3 is uniform, in accord with electron microscopy studies of contractile ring F-actin which show a relatively even distribution in both *S. pombe* and animal cells (*Kamasaki et al., 2007*; *Henson et al., 2017*). In contrast to their membrane-proximal termini, the opposite termini of Myo2 and Cdc15 appear quite uniform. A long flexible region between Cdc15's F-BAR and SH3 domains likely acts to eliminate any F-BAR-oligomerization-mediated clustering once the SH3 domain is reached. The even distribution of Myo2's head domain suggests a generally even binding of actin throughout the ring. This distribution is in contrast to myosin-II 'stacks' formed in animal cells, where antiparallel myosin units are assembled into large-scale clusters parallel to F-actin filaments (*Fenix et al., 2016*; *Henson et al., 2017*).

## Building the ring bottom up

Together with prior knowledge of when proteins are recruited to the ring, the spatial architecture of the contractile ring determined here suggests possible mechanics of ring formation. In fact, the different layers seen in our analysis generally correlate with the order of assembly of their components. Of the earliest ring components which are recruited to precursor nodes (Mid1, Rng2, Myo2, Cdc15, and Cdc12) (*Wu et al., 2006*; *Laporte et al., 2011*), 4/5 contain termini present within the proximal membrane layer (0–80 nm), and 2/5 contain direct membrane binding domains. The precise order of assembly of Rng2, Myo2, and Cdc12 does not appear to matter for ring formation (*Tao et al., 2014*), but it seems likely that the initial setup of the ring involves the construction of a membrane-anchored scaffold. F-actin is formed de novo at nodes and the ring by Cdc12 (*Pelham and Chang, 2002*) as well as incorporated from longitudinal actin cables (*Huang et al., 2012*). F-actin, though turning over quickly in the ring (*Henson et al., 2017*), may be principally held in its elevated location by the tension from the motor activity of myosins. Circumferential tension due to myosin may produce force on F-actin and other components toward the center of the cell, as in 'purse-string' models of ring function (*Henson et al., 2017*). Once a contiguous ring is formed, many additional components are recruited. Components that join the ring later are generally present within the intermediate layer. Many of these components, therefore, may 'fill in' between the membrane-bound proximal layer and F-actin to form a robust structure that withstands the forces of constriction,

couples ring constriction with other events of mitosis, and connects the ring to cell wall formation that occurs coincidentally.

## Materials and methods

### Yeast methods and strain construction

*S. pombe* strains (*Supplementary file 1*) were grown in yeast extract (YE) media at 25°C, with the exception of *pnmt1-Acyl-mMaple3* (*Figure 1—figure supplement 1B*) and *nmt1-Acyl-GFP* (*Figure 5A*) which were grown in Edinburgh minimal medium (EMM) lacking thiamine to induce expression (*Basi et al., 1993*). *S. pombe* transformations were performed with a lithium acetate method (*Keeney and Boeke, 1994*). mMaple3 (*Wang et al., 2014*) was cloned into a pFA6a vector at AscI and PacI sites. C-terminal fluorophore endogenous fusion strains were created by transforming a pFA6a integration cassette amplified with gene specific primers for insertion at the 3' end of open reading frames (*Bähler et al., 1998*). N-terminal mMaple3 or GFP fusion proteins were created with two methods. For nonessential genes (Cyk3, Imp2, Mid1, Pom1, Pxl1, Rga7, and Spa2), constructs containing (1) a 500 bp 5' flank, (2) mMaple3 or GFP with a GGGGSGGGGSG C-terminal linker, (3) the coding sequence, and (4) a 500 bp 3' flank were assembled into a pSK or pIRT2 vector (*Gibson et al., 2009*). These cassettes were amplified from their 5' and 3' flanks, transformed into $ura4^+$ deletion strains of the targeted gene, and selected for $ura4^+$ loss and integration by resistance to 5-FOA. For essential genes (Cdc12, Cdc15, Myo2, Rng2, and Rgf3), constructs containing (1) a 500 bp 5' flank, (2) mMaple3 or GFP with a GGGGSGGGGSG C-terminal flexible linker, (3) the coding sequence, (4) a Kan$^R$ cassette from pFA6a, and (5) a 500 bp 3' flank were assembled into a pSK vector. These cassettes were digested before and after their 5' and 3' flanks and transformed into wildtype *S. pombe*. G418 resistant transformants were screened for correct integration by PCR. *pAct-LifeAct-mMaple3* was constructed to be identical to the previously described *pAct-LifeAct-GFP* (*Huang et al., 2012*). A 1 kb promotor from *act1*, the LifeAct peptide (MGVADLIKKFESISKE) (*Riedl et al., 2008*), a short GGPGG linker, and mMaple3 were assembled into a pJK148 plasmid which was subsequently digested by NruI and integrated into the *leu1-32* locus (*Keeney and Boeke, 1994*). Acyl-mMaple3 was constructed with the N-terminal acyl sequence from Gpa1 (MGCMSSK YADTSGGEV) and mMaple3 in a pREP1 vector (*Maundrell, 1993*). All strains were confirmed by PCR and sequencing.

### Sample preparation for fPALM imaging

To enrich for cells undergoing cytokinesis for imaging, newly separated short cells containing mMaple3-tagged proteins were isolated from a 7–30% lactose gradient, grown for 80 min in YE, labeled with 0.5 mM mCling-ATTO647N (*Revelo et al., 2014*) (Synaptic Systems, Goettingen, Germany) for 10 min, and fixed. The resulting fixed sample was enriched for cells containing fully formed but unconstricted contractile rings. Cells were fixed with 3.7% paraformaldehyde directly in media for 20 min and subsequently washed 6X with PBS. Staining of actin with phalloidin was performed on fixed cells with 3.3 µM Phalloidin-Alexa488 in PBS + 0.1% NP-40 for 30 min, followed by 3 washes with PBS. Fixed and washed cells were resuspended in a small volume of PBS containing 1:100 diluted 80 nm gold particles (Microspheres-Nanospheres, Cold Spring, NY) previously sonicated for 1 hr to disperse clusters. The cell and gold particle suspension was mounted on a 3% agarose pad to prevent cell drifting during imaging.

### fPALM imaging

fPALM imaging (*Figures 1–4* and *6*) was performed on a Nikon dSTORM 4.0 system which included a Nikon Eclipse Ti microscope, 405, 488, 561, and 647 nm solid state lasers, a Hamamatsu ORCA-Flash4.0 camera, and a 100X CFI HP Apochromat TIRF 1.49NA objective with a 1.5X tubelens (resulting in 110 nm pixels). Imaging was performed at a 0° laser angle ('straight through') focused in Z at the center of cells for *Figures 1–4* and the top of cells for *Figure 6*. Z drift was minimized using a Nikon Perfect Focus system. The mMaple3 channel was imaged with simultaneous 0.5% 405 nm activation and 7.5% 561 nm excitation lasers, filtered through a polychroic mirror (ZT405/488/561/647rpc, Chroma, Bellows Falls, VT) and emission filter (ET585/65 m, Chroma), and captured with 30 ms exposures over 10–20 k frames. The ATTO647N channel was imaged with simultaneous 0.2%

405 nm activation and 2% 647 nm excitation lasers, filtered through a polychroic mirror (ZT405/488/ 561/647rpc, Chroma) and emission filter (ET705/75 m, Chroma), and captured with 10 ms exposures over 15 k frames. The Alexa488 channel for Phalloidin-Alexa488 was imaged with simultaneous 0.2% 405 nm activation and 2% 488 nm excitation lasers, filtered through a polychroic mirror (ZT405/488/ 561/647rpc, Chroma) and emission filter (ET525/50 m, Chroma), and captured with 10 ms exposures over 15 k frames. Laser powers and exposure times were optimized for single photoactivated localizations per ring in each frame.

## fPALM analysis

fPALM images were analyzed using the ImageJ plugin ThunderSTORM (*Ovesný et al., 2014*). Images were pre-filtered with a wavelet B-spline filter and molecules were approximately localized with the 8-neighborhood local maximum method. Molecules were sub-pixelly localized using the elliptical Gaussian weighted least squares method to identify precise lateral and Z positions. A Z-calibration was performed with 0.1 µm Tetraspek beads (ThermoFisher) for accurate Z positioning. Axial drift, though minimal with this microscope setup, was corrected in post-processing by tracking fiducial gold bead markers as well as cross-correlation analysis. fPALM images are visualized with localizations as normalized Gaussians, where each peak's standard deviation (FWHM) equals its localization uncertainty (*Betzig et al., 2006*).

Processed and aligned 2-color images were restricted to a 500 nm plane through the center of the cells. Fully formed, unconstricted rings were identified visually through the following criteria: (1) a lack of precursor nodes, (2) a fully formed contiguous ring perpendicular to the long cell axis, and (3) a lack of membrane and septum ingression detected in the mCling channel. These criteria place all analyzed rings in a short window of 15–20 min between ring formation and constriction (*Wu et al., 2003*). The edge of the plasma membrane was identified in an unbiased manner using an automated threshold method to determine where the signal from the mCling-ATTO647N channel drops to 5% of its maximum plasma membrane intensity (*Figure 1—figure supplement 1B*). The distance of each individual mMaple3 particle in the contractile ring to this edge was calculated. These data were fit with a Gaussian curve and the distance from the membrane ($d_{center}$) and a vertical width parameter ($\sigma_{vert}$ or FWHM) were determined using R (*Figure 1E*), as performed previously in a study of focal adhesion proteins (*Kanchanawong et al., 2010*). A horizontal width parameter ($\sigma_{width}$) was also calculated to describe the FWHM of ring localizations parallel to the plasma membrane. $d_{center}$, $\sigma_{vert}$, and $\sigma_{width}$ values from multiple rings (see *Figure 1—figure supplement 1* and *Figure 2—figure supplement 1* for n values and other statistics) were plotted using ggplot2 in R.

The contractile ring horizontal width dimension reported in the text (182 ± 26 nm) represents an average of all components' $\sigma_{width}$. The contractile ring maximum extension from the plasma membrane figure reported in the text (293 ± 64 nm) represents LifeAct-mMaple3's (the most distal component) $d_{center} + \frac{1}{2}(\sigma_{vert})$, approximating the ring's extension into the cytoplasm.

Analysis of local homogeneity in *Figure 6* was performed with downconverted 8-bit images using the 'Texture Analysis' ImageJ plugin (https://imagej.nih.gov/ij/plugins/texture.html). This plugin computes a grey-level co-occurrence matrix for all the pixels in the ROI of a contractile ring $p(i, j)$, and calculates multiple of Haralick's textural features (*Haralick et al., 1973*) from this matrix. We utilized the Inverse Difference Moment (IDM), a measure of local homogeneity:

$$\sum_{i=1}^{N_g} \sum_{j=1}^{N_g} \frac{1}{1 + (i-j)^2} p(i,j)$$

where $N_g$ is the maximum grey value and $p(i, j)$ is the grey level co-occurrence matrix. This analysis was performed specifically in the direction of the ring circumferential axis.

## FRET imaging

FRET imaging (*Figure 5*) was performed on a Perkin Elmer Ultraview Vox spinning disk system equipped with a Zeiss Axio Observer microscope, 488 and 561 nm solid state lasers with a PhotoKinesis bleaching module, a Yokogawa CSU-X1 spinning disk, a 63X C-Apochromat objective, and a Hamamatsu ImageEM C9100-13 EMCCD camera. An acceptor photobleaching method was employed for FRET imaging in live cells. The mCherry-Cdc15 or Cdc15-mCherry fluorophores were bleached throughout the cells with 100% 561 nm laser power for 20 cycles. 10 frames of single Z

slices in the donor GFP channel were acquired pre- and post-bleach. FRET percentages were calculated by first correcting for background and photobleaching over the 10 pre- and post-bleach frames, and subsequently calculating the percentage increase in GFP donor fluorescence at contractile ring ROIs of a consistent size. Statistical tests in *Figure 5A–B* and *Figure 5—figure supplement 1B* and S5B are ANOVA tests versus Rlc1-GFP a negative control which does not FRET with either mCherry-Cdc15 or Cdc15-mCherry, with uncorrected p values reported in *Figure 5—figure supplement 1A–B*. Statistical tests in *Figure 5—figure supplement 1C* were Fisher's exact tests between the mCherry-Cdc15 and Cdc15-mCherry conditions.

## Acknowledgements

We thank members of the Gould lab for critical review of the manuscript, Xiaowei Zhuang for the mMaple3 constructs, and Chunkai Zhou for FRET microscope assistance. fPALM imaging was performed in the Vanderbilt Cell Imaging Shared Resource and FRET imaging was performed in the Stowers Institute Microscopy Center.

## Additional information

### Funding

| Funder | Grant reference number | Author |
| --- | --- | --- |
| National Institutes of Health | Research Grant | Rong Li<br>Kathleen L Gould |
| American Heart Association | Graduate Student Fellowship | Nathan A McDonald |

The funders had no role in study design, data collection and interpretation, or the decision to submit the work for publication.

### Author contributions

Nathan A McDonald, Conceptualization, Data curation, Formal analysis, Investigation, Visualization, Methodology, Writing—original draft, Writing—review and editing; Abigail L Lind, Formal analysis, Validation; Sarah E Smith, Resources, Methodology; Rong Li, Resources, Supervision, Methodology; Kathleen L Gould, Conceptualization, Resources, Formal analysis, Supervision, Funding acquisition, Writing—original draft, Project administration, Writing—review and editing

### Author ORCIDs

Nathan A McDonald  http://orcid.org/0000-0003-2716-3881
Rong Li  http://orcid.org/0000-0002-0540-6566
Kathleen L Gould  http://orcid.org/0000-0002-3810-4070

### Decision letter and Author response

Decision letter https://doi.org/10.7554/eLife.28865.019
Author response https://doi.org/10.7554/eLife.28865.020

## Additional files

### Supplementary files

• Supplementary file 1. *S. pombe* strains used in this study.
DOI: https://doi.org/10.7554/eLife.28865.017

• Transparent reporting form
DOI: https://doi.org/10.7554/eLife.28865.018

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
