## [Decision Letter]

Thank you for submitting your article "Nanoscale architecture of the *Schizosaccharomyces pombeS. pombe* contractile ring" for consideration by *eLife*. Your article has been reviewed by three peer reviewers, one of whom is a member of our Board of Reviewing Editors and the evaluation has been overseen by Vivek Malhotra as the Senior Editor. The reviewers have opted to remain anonymous.

The referees were impressed with the use of super-resolution approach to understand organization of the cytokinetic apparatus in fission yeast, which has become an excellent system to study cytokinesis. The reviewers have discussed the reviews with one another and the Reviewing Editor has drafted this decision to help you prepare a revised submission. We hope you will be able to submit the revised version within two months.

We request that you address the following issues (essential revisions), but you might also want to consider other points in the referee comments (nearly all of which are reasonable) in preparing your revisions.

Essential revisions:

First, the accuracy of your estimate of the location of cell membranes was called into question in points 3 and 4 of referee 1 and point 4 of referee 2. Please address these with alternative approaches suggested by the referees, or other approaches that may be available to you.

Second, and linked to the above point, it will be important to generate some double-tagged strains (e.g. Mid–mMaple3 Cdc15–mMaple3), and then to test for two separate gaussian peaks. If they showed 3-4 examples of double peaks for combined tags, then this would address these concerns (point 5 of referee 1).

Finally, please also address why the cell cycle stages you describe are reliable considering the caveats in the synchrony used.

Reviewer #1:

By using the super resolution localization microscopy technique fPALM, McDonald et al., measured the distances between cell membrane and various cytokinetic ring proteins in the fixed *S. pombe* cells. The authors further validated the proximity of the F–BAR protein Cdc15 to its interacting proteins using FRET technique. In addition, the authors also measured the distances of several regulatory proteins such as kinases and GTPases that participate in cytokinesis relative to cell membrane. Together, the authors provided a 2–dimensional architecture of the cytokinetic ring at the nanometer scale.

The cytokinetic ring is a complex contractile machinery, and how the ring proteins are organized to collectively executing cytokinesis is an important but not yet well studied question in the field. The question addressed by McDonald et al., is important towards the understanding of cytokinetic ring function, and fits the scope of *eLife*. However, due to the technical caveats particularly the ambiguity of the cytokinesis stages and the labelling of cell membrane (major comments below), I am not very convinced by the accuracy of their findings. Since the relative molecular distances of ring proteins are at the nanometer scale, a good definition of cell membrane and the cytokinesis stages is essential to this study. Thus, these concerns should be addressed before further consideration for *eLife*.

1) The authors mentioned that they collected synchronized cells for their fPALM experiments (subsection “Sample preparation for fPALM imaging”). However, there is no supporting evidence for a good cell synchronization. The cells shown in Figure 1 do not seem to be at the same cell cycle stages (cells with different ring widths could be observed). Since the localizations of ring proteins may vary in early assembled rings, in matured rings, or in rings ready for constriction, a reliable indicator of cytokinesis stages will be required when performing the fPALM study. It would be useful to include mitotic markers such as SPB distance or spindle length for more precise indexing of cytokinesis progression.

2) The authors only map the ring protein localization at one single time point (+90 minutes after synchronization). To substantiate their findings, the relative localizations of several key proteins such as Cdc15, F-actin, Myo2 at different stages of cytokinesis (ready for constriction vs. constricting) should be profiled at higher temporal resolution.

3) A reliable cell membrane marker is critical to the accuracy of measurements in this study. It is not entirely clear if mCling is an appropriate marker for cell membrane in *S. pombe* as the dye binds to cell wall strongly (>100 nm thick structure in Figure 1—figure supplement 1). How the authors be certain that the cell membrane was indeed labelled by mCling? The punctate staining of mCling in Figure 1 also suggests that not all membrane/cell wall could be stained. The different sizes of the mCling 'blobs' in Figure 2 and Figure 3 suggest uneven binding to cell wall, which may result in the bias and inaccurate cell membrane edge determination used by authors.

4) It is unclear how the internal edge of cell membrane in the study is determined. Was it mathematically defined for every single measurement or was it 'visually' defined? This is especially important as precision of fPALM nanoscale mapping of ring proteins strongly dependent on precision of localization of plasma membrane. A detailed description of how this was done is required in the Materials and methods section.

5) To substantiate the relative distances of ring proteins, the authors could express and resolve two mMapple3 tagged proteins such as Mid1–mMapple3/mMapple3–Cdc15 (proteins close to cell membrane) and mMapple3–Myo2/LifeAct–mMapple3 (actin rich domain) with fPALM in the same cells. Ideally, two Gaussian peaks could be detected compared to cells expressing only one kind of tagged proteins.

Reviewer #2:

Constriction of the plasma membrane during cytokinesis depends on acto-myosin based contractile ring at the cell centre. Although we know the components of CR, lack of information regarding the nature of molecular arrangement and interaction has severely limited our understanding of the process. In S.pombe, a conserved actomyosin ring of 400nm thickness composed of 130 proteins drives constriction. Such molecular complexity is beyond the scope of diffraction limited optics. This paper (along with a recent work from Pollard lab) attempts at elucidating the structure of actomyosin ring using super-resolution microscopy. The authors suggest a stratified structure of membrane binding components proximal to the membrane and force generators distal to it. While both the scope of the study and its methodical execution deserves much credit, the manuscript suffers from lack of clarity and adequate discussion.

1) The authors show that the components of the Contractile rings are arranged in three distinct zones relative to plasma membrane in the contractile ring prior to ring constriction. My major concern is with lack of temporal information in this manuscript. Is this organization (protein-map) is altered during constriction? Do distribution of some components of Proximal zone change cluster-like to homogenous as the rings constrict or do they retain their conformation?

2) The authors rightly argue that establishment of ultrastructural detail of the actomyosin ring is important to understand their function. However, they do not discuss the important implications of their observation. Myo2 tail is localised to membrane proximal zone where as its head domain is more distal and uniform. Does this mean that type 2 myosin is not arranged as a bipolar filament in contractile ring? Does this distribution change when the rings undergo constriction?

3 "At similar intermediate distances from the membrane, we find the tail domains of two other myosins: the non-essential myosin–II, Myp2, as well as a myosin–V, Myo51."

The manuscript doesn't provide any information on the head of these two motors. Given the recent implications of myosin motors in force generation and mutually exclusive localisation in the contractile ring, I think it may be worthwhile to investigate relative positioning of myo2 and myp2 (Laplante et al., 2015).

4) fPALM strategy for determining the spatial organization of contractile ring components

"The plasma membrane was defined as the edge of the mCling–ATTO647N signal, and the distance from this line was calculated for each individual localization in a contractile ring side-view "spot".

I am unable to fathom the accuracy of the plasma membrane position. If a line was drawn at the edge of the signal of plasma membrane manually it leaves much scope for errors. Signals such as the ones in Figure 1 and Figure 1—figure supplement 1, clearly indicates a diffuse signal near the edge. How did the authors trace such a signal? Was it based on any particular threshold? Given, that this is the point of reference for the entire study, I hope the authors illustrate exactly how they performed their analysis.

5) "We imaged synchronized cells that contained fully formed contractile rings (e.g. no precursor nodes remaining) that had not begun constriction (scored by membrane ingression and septum formation)".

The synchronisation achieved in lactose gradient selection of G1 cells, is not mentioned in the text. Was there any particular reason for G1 synchronisation and not G2?

In the absence of a cell-cycle marker and given the maturation period of S.pombe rings, it seems the authors imaged rings at least 20 minutes apart in their maturation. Certainly, this is enough scope for molecular rearrangements. Since incorporating a third fluorophore is challenging with the current set up, could the authors at least investigate simultaneous position and orientation of two ring components with each other and suggest they do not differ significantly from single labelling?

6) "Therefore, we performed additional fPALM imaging at the top plane of cells (Figure 5) to capture a lateral view of the contractile ring and assess the organization of a subset of contractile ring components".

I am unable to follow how the authors determine "top plane of cells". In contrast to the medial plane where the contractile ring serves as a reliable landmark with two points of maximal separation, "top of the cell" has no such landmarks. How does the authors reliably find the top focal plane? It would be useful if the authors can give an approximation of the circumferential length the 500nm Z section encompasses.

Reviewer #3:

In this study, McDonald et al., used fPALM to map the organization of proteins within the cytokinetic ring. Specifically, they measured the average distance of 29 ring proteins from the plasma membrane, revealing vertical zones of enrichment that make sense given the known functions and interactions of these proteins. For several factors, these positions are further confirmed by FRET. This work follows two previous studies that mapped a more limited set of proteins in cytokinetic ring precursors by SHREC (Laporte, 2011) and fPALM (Laplante, 2016). This McDonald et al., study is an impressive body of work that provides the most detailed account of protein organization in the cytokinetic ring, at least to my knowledge. It will be a reference point for future mechanistic studies. My main concern is that the work is entirely descriptive, it is strictly a localization study. Ideally, one would like to see the function of these localization results tested – what aspects of this vertical organization are important for cytokinesis? As such, it is more of a "resource" than a "research article." Nonetheless, it represents a major step forward for the field, and is likely to draw broad interest. I have a couple of comments that the authors might address to strengthen the manuscript:

Figure 1: Are all proteins distributed in the same width within the ring? The authors measure distance from the membrane and vertical distribution (i.e. spread in distance away from the membrane) for each protein. However, it would be interesting to know if all proteins are distributed along the same horizontal width or not. Presumably, these data are readily available from the images, and the authors state in the text an overall ring width of 181nm, but I did not see how this number was calculated. Measurement for each component would further refine the molecular architecture of the ring by adding additional confinement for each protein.

Figure 5: The data suggest that ring proteins are clustered near the membrane, but not clustered in the more interior strata. Particularly striking are examples like Myo2 and Cdc15, where the cortical domains are clustered, but their 'inner' domains are unclustered. As the authors know, the Pollard lab published fPALM data showing that both termini of these same constructs are clustered in the ring. In fact, all 6 constructs (4 proteins, 2 of which were tagged at each terminus) examined in the Pollard study were clustered. One difference between the two studies is that the Pollard lab imaged live cells, while the McDonald study imaged fixed cells, raising the question of whether fixation affected clustering (there are other difference and possibilities as well). I wonder if the authors could address this discrepancy by imaging lateral organization in live cells. They could also consider imaging more examples of cortical versus interior proteins, to support further this general trend of clustered at the membrane but diffuse at the interior. Finally, measurements of horizontal spread in middle focal planes (previous comment) might show tight clusters at the cortex with broad spread at the interior.

---

## [Author Response]

Essential revisions:First, the accuracy of your estimate of the location of cell membranes was called into question in points 3 and 4 of referee 1 and point 4 of referee 2. Please address these with alternative approaches suggested by the referees, or other approaches that may be available to you.

We have addressed this point (see detailed explanations below) by confirming mCling is a genuine membrane marker, and implementing an unbiased automated method to define the edge of the membrane. Importantly, using this method has not changed any of our major conclusions.

Second, and linked to the above point, it will be important to generate some double-tagged strains (e.g. Mid–mMaple3 Cdc15–mMaple3), and then to test for two separate gaussian peaks. If they showed 3-4 examples of double peaks for combined tags, then this would address these concerns (point 5 of referee 1).

We have now imaged and analyzed multiple double tagged strains with different combinations of proximal, intermediate, and distal components. Confirming our individual measurements, we find double tags in the same layer form a single peak, while double tags in different layers form either double peaks or extra-wide single peaks, depending on their distance apart.

These data have been included in a new Figure 4 and Figure 4—figure supplement 1.

Finally, please also address why the cell cycle stages you describe are reliable considering the caveats in the synchrony used.

We unfortunately misled the reviewers regarding the purpose of the lactose gradient. In fact, it was utilized simply to *enrich* for cells in a single cell cycle stage so that more cells could be found at the appropriate stage in each field of view. Fully formed, unconstricted rings were subsequently identified visually based on very strict criteria. In other words, only that fraction of rings corresponding to fully formed but unconstructed ring in any single field of view were analyzed. We have clarified the methodology in the manuscript and added examples to illustrate how appropriate rings were identified (Figure 1—figure supplement 1). See our response to reviewer 1 point 1 and reviewer 2 point 5 below for details.

Reviewer #1:By using the super resolution localization microscopy technique fPALM, McDonald et al., measured the distances between cell membrane and various cytokinetic ring proteins in the fixed S. pombe cells. The authors further validated the proximity of the F–BAR protein Cdc15 to its interacting proteins using FRET technique. In addition, the authors also measured the distances of several regulatory proteins such as kinases and GTPases that participate in cytokinesis relative to cell membrane. Together, the authors provided a 2–dimensional architecture of the cytokinetic ring at the nanometer scale.The cytokinetic ring is a complex contractile machinery, and how the ring proteins are organized to collectively executing cytokinesis is an important but not yet well studied question in the field. The question addressed by McDonald et al., is important towards the understanding of cytokinetic ring function, and fits the scope of eLife. However, due to the technical caveats particularly the ambiguity of the cytokinesis stages and the labelling of cell membrane (major comments below), I am not very convinced by the accuracy of their findings. Since the relative molecular distances of ring proteins are at the nanometer scale, a good definition of cell membrane and the cytokinesis stages is essential to this study. Thus, these concerns should be addressed before further consideration for eLife.1) The authors mentioned that they collected synchronized cells for their fPALM experiments (subsection “Sample preparation for fPALM imaging”). However, there is no supporting evidence for a good cell synchronization. The cells shown in Figure 1 do not seem to be at the same cell cycle stages (cells with different ring widths could be observed). Since the localizations of ring proteins may vary in early assembled rings, in matured rings, or in rings ready for constriction, a reliable indicator of cytokinesis stages will be required when performing the fPALM study. It would be useful to include mitotic markers such as SPB distance or spindle length for more precise indexing of cytokinesis progression.

This comment, also raised by reviewer 2, reflects a misunderstanding of the purpose of the lactose gradient methodology in our study. We apologize for the confusion. The lactose gradient was used to simply *enrich* for cells in the same cell cycle phase but we did not analyze all cells obtained. This step simply allowed us to decrease the number of experiments required to image a sufficient number of cells at the appropriate stage in cytokinesis. We made no assumption about exact cell cycle stage based on the enrichment protocol.

Rather, cells and rings were individually selected for analysis based on criteria that place them within a short window between full ring formation and the beginning of constriction (which is generally considered 15-20 minutes (Wu and Pollard, 2003)). These visual criteria are: (28) a lack of precursor nodes, meaning the first cytokinesis proteins have condensed into a ring, (18) the presence of a fully formed and tightly concentrated ring absolutely perpendicular to the cell length, and (68) the absence of detectable ring constriction or membrane ingression. We have reworded the methods and main text to clarify our use of the lactose gradient, and added a panel in Figure 1—figure supplement 1 that explicitly lays out the criteria used for selecting mature pre-constriction rings with examples.

It is possible that certain early ring components reorganize during the short period between initial ring formation and ring constriction, which is on the order of a few minutes. One fact to note is that the majority of proteins we have analyzed are not present at the division site in nodes or during the initial stages of ring formation meaning that if they are present in rings, the rings are fully formed and ready to constrict. Furthermore, visualizing mitotic markers such as spindle pole bodies will not allow us to define any more precise temporal window post- anaphase B onset when the majority of proteins are assembled into the ring and constriction. We have gathered data on a large number of rings using our criteria; temporally these ring populations are either all mature for late-arriving proteins, or for some proteins span full assembly through to maturation. Therefore, the data represent an average across this very short time window. That no single protein has a large deviation from its mean and that the deviations from the mean are similar across all proteins suggests significant individual protein rearrangements do not occur.

We believe the criteria we have employed to analyze protein localization exclusively in fully formed but not yet constricting rings achieves excellent temporal resolution and the best accuracy currently technically feasible in fPALM fixed samples.

2) The authors only map the ring protein localization at one single time point (+90 minutes after synchronization). To substantiate their findings, the relative localizations of several key proteins such as Cdc15, F–actin, Myo2 at different stages of cytokinesis (ready for constriction vs. constricting) should be profiled at higher temporal resolution.

In the future we hope to determine if ring architecture changes during constriction. However, we believe this is beyond the scope of the current study. For one, such experiments will require significant additional technical optimization because the mCling dye does not uniformly label the “ingressing” membrane, possibly because new membrane is added primarily from vesicular trafficking and would require much longer labelling with the dye to achieve uniform membrane labeling in the furrow.

3) A reliable cell membrane marker is critical to the accuracy of measurements in this study. It is not entirely clear if mCling is an appropriate marker for cell membrane in S. pombe as the dye binds to cell wall strongly (>100 nm thick structure in Figure 1—figure supplement 1). How the authors be certain that the cell membrane was indeed labelled by mCling? The punctate staining of mCling in Figure 1 also suggests that not all membrane/cell wall could be stained. The different sizes of the mCling 'blobs' in Figure 2 and Figure 3 suggest uneven binding to cell wall, which may result in the bias and inaccurate cell membrane edge determination used by authors.

mCling is currently the only membrane marker compatible with super-resolution imaging and fixation (see its extensive characterization in Revelo et al., 2014). We recognize this membrane dye is not perfect due to its binding to the cell wall, but to our knowledge it is only current option for this analysis.

To support this probe’s use as a membrane marker, we have compared the mCling label with the widely used FM4–64 membrane dye using conventional resolution imaging (Figure 1—figure supplement 1). We find that these two membrane probes colocalize strongly and both are trafficked into intracellular membrane compartments, leading us to conclude mCling is a bona fide membrane marker. Comparing the mCling dye’s localization with a membrane tethered acyl-mMaple3 with fPALM also reveals an exact apposition of these two signals (Figure 1—figure supplement 1), supporting mCling’s use as a membrane marker.

4) It is unclear how the internal edge of cell membrane in the study is determined. Was it mathematically defined for every single measurement or was it 'visually' defined? This is especially important as precision of fPALM nanoscale mapping of ring proteins strongly dependent on precision of localization of plasma membrane. A detailed description of how this was done is required in the Materials and methods section.

We originally identified the internal edge of the membrane visually. However, due to the concern about the manual nature of this approach, we implemented an automated method to identify the membrane edge. This method uses the position where the mCling intensity drops by 95% to mark the edge of the plasma membrane in an unbiased manner. All the data has been re-analyzed using this automated method. Satisfyingly, implementing the new method did not result in any significant changes in d_center_ or σ_vert_ values.

5) To substantiate the relative distances of ring proteins, the authors could express and resolve two mMapple3 tagged proteins such as Mid1-mMapple3/mMapple3-Cdc15 (proteins close to cell membrane) and mMapple3-Myo2/LifeAct-mMapple3 (actin rich domain) with fPALM in the same cells. Ideally, two Gaussian peaks could be detected compared to cells expressing only one kind of tagged proteins.

We thank the reviewer for this excellent suggestion. We have performed this analysis on multiple pairs of proteins found in similar and distinct layers. As predicted, we find double tags in the same layer form single peaks. Supporting our individual measurements of proteins in different layers, we find double tags in different layers visualize as extra-wide single or resolvable double peaks, depending on their distance apart. We present examples of these double tags compared to their individual components in a new Figure 4 and Figure 4—figure supplement 1. We also compare the σ_vert_ values (a measure of the vertical width of the gaussians) of the double tags to their individual counterparts to show a “widening” of peaks when the components are far apart in a new Figure 4. These data have strengthened our conclusions about the spatial separation of ring components relative to the plasma membrane.

Reviewer #2:Constriction of the plasma membrane during cytokinesis depends on acto-myosin based contractile ring at the cell centre. Although we know the components of CR, lack of information regarding the nature of molecular arrangement and interaction has severely limited our understanding of the process. In S.pombe, a conserved actomyosin ring of 400nm thickness composed of 130 proteins drives constriction. Such molecular complexity is beyond the scope of diffraction limited optics. This paper (along with a recent work from Pollard lab) attempts at elucidating the structure of actomyosin ring using super-resolution microscopy. The authors suggest a stratified structure of membrane binding components proximal to the membrane and force generators distal to it. While both the scope of the study and its methodical execution deserves much credit, the manuscript suffers from lack of clarity and adequate discussion.1) The authors show that the components of the Contractile rings are arranged in three distinct zones relative to plasma membrane in the contractile ring prior to ring constriction. My major concern is with lack of temporal information in this manuscript. Is this organization (protein-map) is altered during constriction? Do distribution of some components of Proximal zone change cluster-like to homogenous as the rings constrict or do they retain their conformation?

These are very interesting points and we intend to pursue the question of possible rearrangements during constriction in the future. As mentioned in our response to reviewer 1 point 2 above, technical challenges have so far limited our analysis to pre-constriction contractile rings. We must develop new methods to interrogate the constricting plasma membrane reliably.

2) The authors rightly argue that establishment of ultrastructural detail of the actomyosin ring is important to understand their function. However, they do not discuss the important implications of their observation. Myo2 tail is localised to membrane proximal zone where as its head domain is more distal and uniform. Does this mean that type 2 myosin is not arranged as a bipolar filament in contractile ring? Does this distribution change when the rings undergo constriction?

Indeed, Myo2 is not arranged as a bipolar filament in nodes or the contractile ring and this has been noted previously (Laporte et al., 2011; Laplante et al., 2016; Pollard et al., 2017). We have added a discussion point noting this difference from mammalian cells (subsection “Yeast methods and strain construction*”*). As described above, examining constricting rings for any changes is beyond the scope of this study.

3 "At similar intermediate distances from the membrane, we find the tail domains of two other myosins: the non–essential myosin–II, Myp2, as well as a myosin-V, Myo51."The manuscript doesn't provide any information on the head of these two motors. Given the recent implications of myosin motors in force generation and mutually exclusive localisation in the contractile ring, I think it may be worthwhile to investigate relative positioning of myo2 and myp2 (Laplante et al., 2015).

We have created the endogenous N–terminal mMaple3 fusions of Myo51 and Myp2 and added analysis of these motor domains. As expected, we found these domains in the same layer as F-actin where they can perform their motor activities, distal to the plasma membrane. In the Laplante et al., study, using conventional microscopy it was concluded that Myo2 and Myp2 co-localize before constriction but not during constriction. We agree with the findings that the motor domains co-localize prior to constriction.

4) fPALM strategy for determining the spatial organization of contractile ring components."The plasma membrane was defined as the edge of the mCling–ATTO647N signal, and the distance from this line was calculated for each individual localization in a contractile ring side-view "spot".I am unable to fathom the accuracy of the plasma membrane position. If a line was drawn at the edge of the signal of plasma membrane manually it leaves much scope for errors. Signals such as the ones in Figure 1 and sup Figure 1, clearly indicates a diffuse signal near the edge. How did the authors trace such a signal? Was it based on any particular threshold? Given, that this is the point of reference for the entire study, I hope the authors illustrate exactly how they performed their analysis.

We originally identified the internal edge of the membrane visually. However, due to the concern about the manual nature of this approach, we implemented an automated method to identify the membrane edge. This method uses the position where the mCling intensity drops by 95% to mark the edge of the plasma membrane in an unbiased manner. All the data has been re-analyzed using this automated method. Satisfyingly, using the new method did not result in any significant changes in d_center_ and σ_vert_ values.

5) "We imaged synchronized cells that contained fully formed contractile rings (e.g. no precursor nodes remaining) that had not begun constriction (scored by membrane ingression and septum formation)".The synchronisation achieved in lactose gradient selection of G1 cells, is not mentioned in the text. Was there any particular reason for G1 synchronisation and not G2?In the absence of a cell-cycle marker and given the maturation period of S.pombe rings, it seems the authors imaged rings at least 20 minutes apart in their maturation. Certainly, this is enough scope for molecular rearrangements. Since incorporating a third fluorophore is challenging with the current set up, could the authors at least investigate simultaneous position and orientation of two ring components with each other and suggest they do not differ significantly from single labelling?

Please see response to reviewer 1, point 1 above. We again apologize for the wording in our original text.

6) "Therefore, we performed additional fPALM imaging at the top plane of cells (Figure 5) to capture a lateral view of the contractile ring and assess the organization of a subset of contractile ring components".I am unable to follow how the authors determine "top plane of cells". In contrast to the medial plane where the contractile ring serves as a reliable landmark with two points of maximal separation, "top of the cell" has no such landmarks. How does the authors reliably find the top focal plane? It would be useful if the authors can give an approximation of the circumferential length the 500nm Z section encompasses.

The top focal plane was determined by the presence of a ring “arc”, which is an equally reliable landmark as the two ring “spots” seen in the middle plane. Assuming a 3.5µm cell diameter, a 500nm Z section centered at the top of the cell should reveal at maximum a 1.9µm contractile ring “arc” at the top of the cell. This is very similar to the size of ring arcs we analyzed.

Reviewer #3:In this study, McDonald et al., used fPALM to map the organization of proteins within the cytokinetic ring. Specifically, they measured the average distance of 29 ring proteins from the plasma membrane, revealing vertical zones of enrichment that make sense given the known functions and interactions of these proteins. For several factors, these positions are further confirmed by FRET. This work follows two previous studies that mapped a more limited set of proteins in cytokinetic ring precursors by SHREC (Laporte, 2011) and fPALM (Laplante, 2016). This McDonald et al., study is an impressive body of work that provides the most detailed account of protein organization in the cytokinetic ring, at least to my knowledge. It will be a reference point for future mechanistic studies. My main concern is that the work is entirely descriptive, it is strictly a localization study. Ideally, one would like to see the function of these localization results tested – what aspects of this vertical organization are important for cytokinesis? As such, it is more of a "resource" than a "research article." Nonetheless, it represents a major step forward for the field, and is likely to draw broad interest. I have a couple of comments that the authors might address to strengthen the manuscript:Figure 1: Are all proteins distributed in the same width within the ring? The authors measure distance from the membrane and vertical distribution (i.e. spread in distance away from the membrane) for each protein. However, it would be interesting to know if all proteins are distributed along the same horizontal width or not. Presumably, these data are readily available from the images, and the authors state in the text an overall ring width of 181nm, but I did not see how this number was calculated. Measurement for each component would further refine the molecular architecture of the ring by adding additional confinement for each protein.

We thank the reviewer for the suggestion, and have added this recommended analysis of ring width (σ_width_) to Figure 2 and Figure 3—figure supplement 1A. Widths appear to vary between proteins, without much correlation with distance from the membrane. We have added a discussion point to this effect (subsection “Structural layers within the contractile ring”).

We also now report a more accurate ring with value in the text (182nm) that represents an average of the 29 proteins’ widths.

Figure 5: The data suggest that ring proteins are clustered near the membrane, but not clustered in the more interior strata. Particularly striking are examples like Myo2 and Cdc15, where the cortical domains are clustered, but their 'inner' domains are unclustered. As the authors know, the Pollard lab published fPALM data showing that both termini of these same constructs are clustered in the ring. In fact, all 6 constructs (4 proteins, 2 of which were tagged at each terminus) examined in the Pollard study were clustered. One difference between the two studies is that the Pollard lab imaged live cells, while the McDonald study imaged fixed cells, raising the question of whether fixation affected clustering (there are other difference and possibilities as well). I wonder if the authors could address this discrepancy by imaging lateral organization in live cells. They could also consider imaging more examples of cortical versus interior proteins, to support further this general trend of clustered at the membrane but diffuse at the interior. Finally, measurements of horizontal spread in middle focal planes (previous comment) might show tight clusters at the cortex with broad spread at the interior.

We have altered the language of the discussion to include the difference in methods to qualify our conclusions. It is unfortunately not possible to image circumferential organization in live cells with our microscope setup, as image acquisition takes 10+ minutes in each channel.

There are certainly caveats to the use of fixed cells, which is why we minimized the extent and length of fixation and used FRET methodology in live cells to corroborate our findings. Still, we cannot rule out that the evenness of signal in the interior of the ring is somehow affected by fixation. However, there are also significant caveats with the use of live cells for fPALM. Though fixed cells are disadvantageous due to loss of the dynamics of the cytokinesis process, fixing is critical to allow superresolution fluorophores to be exhaustively photoconverted and excited over multiple minutes. Uneven conversion and excitation becomes quite possible with the fast and short acquisitions necessary in live cells, perhaps biasing the results of image analysis towards punctate patterns. We also note that the methodology reported in the Pollard study lacked any fiducial marking for drift correction which could also contribute to the development of punctate images.

To expand this analysis as suggested, we imaged and analyzed more ring components in the circumferential dimension. In the proximal layer, we added septin Spn3 and found a uniform distribution, and in the intermediate layer we have added Fic1 and the tail of Myp2, finding that both are uniform. The trend remains consistent that intermediate and distal components are uniformly distributed, while certain proximal components (generally those that were present in precursor nodes) cluster.